# FIGURO - INTRINSIC DIMENSION ESTIMATION FOR MULTI-MODAL DATA

## ABSTRACT

A fundamental challenge in representation learning is determining the complexity, or the *Intrinsic Dimension (ID)*, of the data. This becomes especially difficult in the multi-modal setting when trying to learn disentangled subspaces for shared and private (modality-specific) information. Existing ID estimation techniques are ill-suited for this task, as they are either static and uni-modal or, in the case of state-of-the-art contrastive methods, adapt only to the shared ID implicitly. This leaves a critical gap for a method that can estimate the complete ID structure of multi-modal data. We introduce Fidelity-Guided Rank Optimization (FiGuRO), a framework for learning the IDs of uni- and multi-modal data. FiGuRO learns the dimensions of low-rank projections using truncated singular value decomposition and an algorithm that determines *when* to reduce or increase dimensionalities and in *which* latent spaces. We demonstrate that FiGuRO outperforms other ID estimation techniques and is more robust to hyperparameter changes. In the multi-modal setting, FiGuRO successfully decomposes shared and modality-specific information and captures differences between scales of IDs and varying ratios between the subspaces on simulations and real datasets. Our work provides a quantitative framework for assessing the shared and private informational contributions of multi-modal data. This helps construct more interpretable models and can guide strategic and efficient data collection in fields like biology and medicine.

## 1 INTRODUCTION

Representation learning aims to find low-dimensional representations able to describe complex, high-dimensional data with minimal information loss (Vincent et al., 2008). It is deeply rooted in the Manifold Hypothesis, which posits that real-world data, despite being embedded in a high-dimensional ambient space, is concentrated near a low-dimensional Manifold (Fefferman et al., 2016). Getting close to the Manifold and thus optimal compression requires good estimates of the data complexity, which is a fundamental challenge in representation learning. A critical component of data complexity is its *Intrinsic Dimension (ID)*, defined as the minimum number of variables needed to describe the data without significant information loss. The ID quantifies the true degrees of freedom, and the performance of deep neural networks has been shown to depend on the intrinsic rather than ambient dimension (Nakada & Imaizumi, 2020). The challenge of ID estimation is amplified in the context of multi-modal data. In this setting, the problem transforms from estimating a single ID to a more complex one of disentangling the latent space into shared and modality-specific or "private" subspaces, each with their own unknown ID. Quantifying these distinct IDs is especially important in fields like biology and medicine, where we i) need interpretable models and ii) want to know whether expensive or difficult-to-obtain modalities are relevant (Gliozzo et al., 2025).

Despite its importance, determining the IDs of multi-modal data and its subspaces has remained a fundamental challenge. Traditional ID estimation methods struggle with the curse of dimensionality, often underestimating the true ID and failing to scale to complex, high-dimensional data (Campadelli et al., 2015; Binnie et al., 2025). While more advanced neural network approaches exist, they are often static or designed only for uni-modal data (Bahadur & Paffenroth, 2020; Bonheme & Grzes, 2022; Saha et al., 2025; Potapov & Ali, 2002). In multi-modal learning, most methods either treat latent dimensions as fixed hyperparameters requiring extensive tuning (Bousmalis et al., 2016; Gonzalez-Garcia et al., 2018; Lee & Pavlovic, 2020) or, in the case of state-of-the-art contrastive models, adapt only to a shared ID implicitly as an emergent property (Gui et al., 2025). To the

best of our knowledge, no existing neural network-based approach provides explicit estimates of the distinct IDs of both shared and private subspaces in multi-modal data.

In this work, we introduce a technique for uni- and multi-modal ID estimation called *Fidelity-Guided Rank Optimization (FiGuRO)*. Our approach directly estimates the IDs for each subspace by combining latent rank optimization inspired by adaptive rank reduction (Mounayer et al., 2025), with a measure of the decrease in reconstruction fidelity (distortion). FiGuRO estimates the ID given a controlled level of distortion in a single training pass. We demonstrate FiGuRO's superiority over uni-modal methods when it comes to complex, high-dimensional simulated data. On multi-modal simulations, we show that our method picks up on differences in scale and shared-to-private ratios of ground truth IDs. Lastly, we apply FiGuRO to two real-world multi-modal datasets and validate it through known relationships between the modalities. Our work significantly advances the field of (multi-modal) representation learning by providing both an estimation approach to the content of shared and private information as well as a training paradigm for more interpretable models.

## 2 RELATED WORK

A common line of inquiry of neural network-based ID estimation has focused on making autoencoders aware of the data's geometric structure. Early approaches were often computationally expensive, such as training multiple models with different bottleneck sizes to identify a sharp increase in reconstruction loss, or "loss cliff" (Bahadur & Paffenroth, 2020). More adaptive solutions have been proposed within the VAE (Kingma & Welling, 2022) framework. These include one-time estimation algorithms like FONDUE, which identifies the optimal fixed dimensionality by detecting the collapse of posteriors for irrelevant latent dimensions (Bonheme & Grzes, 2022), and continuous Bayesian methods like ARD-VAE, which adds a hierarchical prior to automatically prune dimensions during training (Saha et al., 2025). However, these approaches inherit the known optimization challenges of the ELBO (Alemi et al., 2018). Rank reduction autoencoders (RRA) present a deterministic counterpart and learn latent representations as matrix decompositions, allowing for dynamic pruning via truncated singular value decomposition (Mounayer et al., 2025). Matrix decomposition has also successfully been used for parameter-efficient fine-tuning frameworks like LoRA (Hu et al., 2021) and LoReFT (Wu et al., 2024), with a recent extension of distributing a fixed rank budget based on explained variance of activations (Paischer et al., 2025).

In the multi-modal setting, methods from classical multi-view data decomposition (Lock et al., 2013; Gaynanova & Li, 2017; Feng et al., 2018; Pandeva & Forré, 2023; Sergazinov et al., 2025) provide estimates for both joint and individual subspaces, but they inherently rely on the assumption of linear mixing of the underlying variables. To the best of our knowledge, a generalizable neural network-based technique enabling disentanglement AND explicitly estimating the complete ID structure is still lacking. Several lines of work address the related challenge of disentangling multi-modal information. In causal representation learning, Sturma et al. (2023) identify shared causal variables from unpaired data, but under restrictive assumptions of linearity and non-Gaussianity. Many multi-modal representation learning approaches propose to fuse modalities by learning joint or aligned representations (Wang et al., 2015; Wu & Goodman, 2018; Shi et al., 2019; Akkus et al., 2023; Sutter et al., 2024). Others have addressed the problem of "modality laziness", which describes a model's tendency to solve a task by relying on the most dominant modality while neglecting the rest (Du et al., 2021). Proposed solutions include uni-modal pretraining (Ismail et al., 2020), gradient balancing (Peng et al., 2022), and alternate modality training (Zhang et al., 2024). Some advanced methods learn separate representations for shared and modality-specific information, but treat the dimensions as hyperparameters (Bousmalis et al., 2016; Gonzalez-Garcia et al., 2018; Lee & Pavlovic, 2020). Recently, it has been proposed that multi-modal contrastive methods like CLIP (Radford et al., 2021) implicitly adapt to the shared intrinsic dimension of the data as an emergent property of optimizing the contrastive loss (Gui et al., 2025). To summarize, related work either focuses on alignment, treats dimensions as fixed hyperparameters, or yields an implicit, uninterpretable estimate of only the shared ID. In contrast, our work provides an explicit approach to estimate the IDs of both shared and modality-specific subspaces, as well as a general training framework for multi-modal models to improve disentanglement and thus interpretability.

## 3 METHODS

Our method, Fidelity-Guided Rank Optimization (FiGuRO), dynamically estimates the intrinsic dimension (ID) of uni- and multi-modal data. FiGuRO learns low-rank projections for each latent subspace inspired by adaptive rank reduction (ARR) (Mounayer et al., 2025), allowing dimensions to be both reduced and increased in order to converge to the ID. Decisions to decrease or increase are guided by principles from Rate-Distortion Theory, using a simple reconstruction fidelity metric.

### 3.1 PRELIMINARIES

**Rate-Distortion Theory and Autoencoders.** Rate-Distortion Theory can be used to describe the trade-off between the complexity of a representation (rate) and its fidelity (distortion). The rate-distortion function, $R(D)$, defines the minimum rate R required to transmit data such that it can be reconstructed with an expected distortion less than or equal to $D$ (Berger, 2003).

$$R(D) = \min_{p(\hat{\boldsymbol{x}}|\boldsymbol{x}) \text{ s.t. } \mathbb{E}[d(\boldsymbol{x},\hat{\boldsymbol{x}})] \leq D} I(\boldsymbol{x};\hat{\boldsymbol{x}}) \tag{1}$$

An autoencoder learns a representation $\boldsymbol{z}$ of a data sample $\boldsymbol{x}$ by training an encoder $f(\boldsymbol{x};\phi)$ with parameters $\phi$ and a decoder $g(\boldsymbol{z};\theta)$ with parameters $\theta$ to reconstruct the input. The reconstruction loss is a direct measure of distortion. The rate is implicitly controlled by the dimensionality of the bottleneck with dimension $k$. The Minimum Description Length principle formalizes this by showing that the description length $L$ of a latent code $\boldsymbol{z} \in \mathbb{R}^k$ is linearly proportional to its dimension (Grünwald, 2007). The bottleneck dimension $k$ is thus a direct, architectural proxy for the rate.

**Multi-modal autoencoders.** A common approach in multi-modal representation learning is to split the latent representation into distinct subspaces learning shared and modality-specific information. For two modalities $\boldsymbol{x}_1$ and $\boldsymbol{x}_2$, latent subspaces are generated either directly or, in deep models, from intermediate representations $\boldsymbol{z}$. We denote these latent subspaces as the shared latent space $\boldsymbol{h}_s$ and private latent spaces $\boldsymbol{h}_1$ and $\boldsymbol{h}_2$. The decoders reconstruct each modality from a combination of its private space and the shared space, where $\oplus$ denotes concatenation.

$$\hat{\boldsymbol{x}}_1 = g_1(\boldsymbol{h}_s \oplus \boldsymbol{h}_1; \theta_1) \quad \text{with} \quad \boldsymbol{h}_1 = f_{1b}(\boldsymbol{x}_1; \phi_{1b}), \ \boldsymbol{h}_s = f_s(\boldsymbol{x}_1, \boldsymbol{x}_2; \phi_s) \tag{2}$$

The formulation for $\hat{\boldsymbol{x}}_2$ is analogous. This architecture encourages $\boldsymbol{h}_s$ to capture modality-invariant information, while $\boldsymbol{h}_1$ and $\boldsymbol{h}_2$ capture modality-specific details.

**Adaptive rank reduction.** ARR is a technique for dynamically reducing the dimensionality of the latent space in a neural network (Mounayer et al., 2025). Instead of learning a full latent matrix $\mathbf{Z} \in \mathbb{R}^{N \times k}$, one can learn its low-rank decomposition. Building on truncated Singular Value Decomposition (SVD), $\mathbf{Z}$ can be approximated by its $k^*$-rank version $\mathbf{Z} \approx \mathbf{Z}^{(k^*)} = \mathbf{U}^{(k^*)}\mathbf{S}^{(k^*)}\mathbf{V}^{T(k^*)}$ where $\mathbf{S}^{(k^*)}$ contains the top $k^* < k$ singular values. This approximation is done by computing the normalized cumulative energies of the singular values $\mathbf{S}$ as $\mathbf{E}_j = \mathbf{S}_j^2 / \sum_j^k \mathbf{S}^2$ and discarding dimensions with an energy below the *energy threshold $\gamma$*.

### 3.2 PROPOSED METHOD: FIDELITY-GUIDED RANK OPTIMIZATION

The core of our contribution is an algorithm that optimizes the dimension of latent subspaces under a user-defined "acceptable" level of lossy compression inspired by ARR and Rate-Distortion Theory. Unlike standard ARR, which applies SVD on latent batches, we decompose the weight matrices similar to LoRA (Hu et al., 2021) to learn a global low-rank structure independent of batch size. Furthermore, we explicitly prune the irrelevant dimensions from the projection, giving us low-dimensional representations. We also enable the increase of ranks if compression has become too lossy. We introduce a low-rank decomposable layer with maximum rank $k_{max}$ into the bottleneck of an arbitrary, sufficiently capable autoencoder with weight matrix $\mathbf{W}$ as follows:

$$\mathbf{W} \approx \mathbf{W}^{(k^*)} = \mathbf{U}^{(k^*)}\mathbf{S}^{(k^*)}\mathbf{V}^{T(k^*)} \tag{3}$$

with $\mathbf{U}^{(k^*)} \in \mathbb{R}^{l \times k^*}$ and $\mathbf{V}^{T(k^*)} \in \mathbb{R}^{k^* \times l}$ as trainable parameters. $l$ denotes the previous hidden dimension in the autoencoder. Rank reduction to $k^*$ is governed by the energy threshold $\gamma$ as in ARR. Following (2) and (3), for a given modality $\boldsymbol{x}_m$, its reconstruction is generated as

$$\hat{\boldsymbol{x}}_m = g_m(\boldsymbol{h}_s \oplus \boldsymbol{h}_m; \theta_m) \quad \text{with} \quad \boldsymbol{h}_m = \mathbf{W}_m^{(k_m^*)}\boldsymbol{z}_m, \ \boldsymbol{h}_s = \mathbf{W}_s^{(k_s^*)}(\boldsymbol{z}_{m=1} \oplus \boldsymbol{z}_{m=2}) \tag{4}$$

where $\mathbf{W}_1^{(k_1^*)}$ and $\mathbf{W}_s^{(k_s^*)}$ are the dynamically adjusted low-rank weight matrices. $\boldsymbol{z}_1 = f_1(\boldsymbol{x}; \phi_1)$ is the uni-modal representation of dimension $l$ that is fed into the decomposed layer. This architecture explicitly estimates the shared dimension ($k_s$) and the private dimension ($k_m$). The ranks are determined by a two-stage process. 1) Reduction is done via cumulative energy reduction as in ARR (using energy threshold $\gamma$, See section 3.1) and dimensions are additionally masked out from the weight matrix to produce low-dimensional representations. Increasing dimensions is achieved by unmasking. 2) The frequency and initiation of these rank reductions are iteratively controlled by Algorithm 1 based on the *distortion budget* $\lambda$. This budget defines how "lossy" we allow the compression to become. After an initial training phase has established a baseline reconstruction quality ("fidelity"), our framework iteratively adjusts the ranks based on $\lambda$. We use the coefficient of determination ($R^2 \in [0,1]$, see A.5) as our fidelity metric, which allows us to define the minimum acceptable fidelity as $R_0^2 - \lambda$ determining when to stop rank reduction, where $R_0^2$ is the initial baseline fidelity. Importantly, we do not need any additional losses, making optimization much simpler and more robust. The full procedure is detailed in Algorithm 1. Our approach can be framed as a greedy algorithm for finding an efficient operating point on the Rate-Distortion curve. It works by minimizing the reconstruction loss given model parameters $\phi$, $\theta$, and rank $k^*$ under the condition that the change in distortion $\Delta\mathbb{E}[d(\mathbf{X}, \hat{\mathbf{X}})]$ is below or equal to the distortion budget $\lambda$.

$$R(D) = \min_{\phi, \theta, k^* \text{ s.t. } \Delta\mathbb{E}[d(\mathbf{X},\hat{\mathbf{X}})] \leq \lambda} \mathcal{L}_{\text{recon}} \tag{5}$$

Fidelity-Guided Rank Optimization (FiGuRO) can give robust estimates of the ID guaranteed to converge under a number of assumptions. Most importantly, we assume the autoencoder architecture is a sufficiently capable function approximator to model the data Manifold and that the chosen distortion metric is a suitable and stable proxy for reconstruction fidelity. Our theoretical guarantees are elaborated in A.1.

## 4 RESULTS

We present our results in the order of experimentation. First, we report on the hyperparameter space and robustness and compare our method to uni-modal ID estimation techniques. Then, we demonstrate FiGuRO's capability to learn IDs of multi-modal shared and private subspaces and to disentangle information on both simulated and real data.

Table 1: **Simulated datasets.** A summary of the datasets we simulated, including the ambient data dimensions, ground truth hidden dimensions for shared and private subspaces, and the number of samples.

|  | A | $\mathbf{B}_s$ "small" | $\mathbf{B}_{i1}$ "imbalanced 1" | $\mathbf{B}_{i2}$ "imbalanced 2" | $\mathbf{B}_l$ "large" | D |
|---|---|---|---|---|---|---|
| Data dimension | 50 | 200, 200 | 200, 200 | 200, 200 | 200, 200 | 20K, 20K |
| True ID |  |  |  |  |  |  |
|     Shared | 5 | 2 | 20 | 2 | 20 | 6 |
|     Modality 1 | – | 3 | 2 | 2 | 20 | 5 |
|     Modality 2 | – | 5 | 2 | 20 | 20 | 6 |
| N samples | 10K | 10K | 10K | 10K | 10K | 30K |

### 4.1 EVALUATION ON SIMULATED DATA

Evaluation of ID estimation techniques requires data with known ground truth dimensions. For this purpose, we generated three simulated datasets of varying complexity in terms of generative process, IDs, and ambient dimensionality. A summary is given in Table 1. Datasets $\mathbf{A}$ and $\mathbf{B}$ present simple simulations which we refer to as "parametric" as they can be generated for a variety of data characteristics such as nonlinearity, underlying distributions, and noise. Uni-modal dataset $\mathbf{A}$ was designed to evaluate hyperparameter sensitivity and robustness to data characteristics. Multi-modal simulation $\mathbf{B}$ was created with four variations that differ in the ratios and scales of the shared and private hidden dimensions in order to test whether our method can pick up on different subspace

---

**Algorithm 1** Fidelity-Guided Rank Optimization

---

1: **Input:** Multi-modal data $\{\mathbf{X}_1, \ldots, \mathbf{X}_M\}$, distortion budget $\lambda$, reduction frequency $\tau$, patience $\pi$, energy threshold $\gamma = 0.01$.
2: **Initialize:** Multi-modal autoencoder with full-rank adaptable layers: $k_{\min} \leftarrow 1, k_{\max} \leftarrow l$
    **Phase 1: Pre-training**
3: Train the full-rank model on reconstruction loss until validation loss stagnates.
4: Compute and store initial fidelity: $R_{0,m}^2 \leftarrow R^2(\mathbf{X}_m, \hat{\mathbf{X}}_m)$ for each modality $m$.
    **Phase 2: Rank Optimization**
5: **for** each epoch $t$ **do**
6:     Train model on reconstruction loss for one epoch.
7:     **for** every $\tau$ epochs **do**
8:         rank_changed $\leftarrow$ False
9:         $M_{decrease} \leftarrow \{\}, M_{increase} \leftarrow \{\}$
10:        **for** each modality $m \in \{1, \ldots, M\}$ **do**
11:           Compute current fidelity $R_{t,m}^2$
12:           **if** $R_{t,m}^2 > R_{0,m}^2 - \lambda$ **then**   Add $m$ to $M_{decrease}$
13:           **else if** $R_{t,m}^2 < R_{0,m}^2 - \lambda$ **then**   Add $m$ to $M_{increase}$
14:           **end if**
15:        **end for**
16:        $K_{decrease} \leftarrow \{\}, K_{increase} \leftarrow \{\}$
17:        **if** $M_{increase} = \{\}$ **then**   $K_{decrease} \leftarrow \{k_s, k_m | m \in M_{decrease}\}$
18:        **else if** $M_{decrease} = \{\}$ **then**   $K_{increase} \leftarrow \{k_s, k_m | m \in M_{increase}\}$
19:        **else**   $K_{decrease} \leftarrow \{k_m | m \in M_{decrease}\}, K_{increase} \leftarrow \{k_m | m \in M_{increase}\}$
20:        **end if**
21:        **for** $k_t$ in $K_{decrease}$ **do**
22:          Decrease rank $k_t$ for $\mathbf{E} < 1 - \gamma$ and set rank_changed $\leftarrow$ True if $k_{t+1} < k_t$
23:        **end for**
24:        **for** $k_t$ in $K_{increase}$ **do**
25:          Increase rank $k_t$ to $k_{t-1}$ and set rank_changed $\leftarrow$ True
26:        **end for**
27:        **if** rank has not changed for more than $\pi$ epochs **then**
28:          **break**
29:        **end if**
30:     **end for**
31: **end for**
32: **Return:** Final ranks $\{k_s^*, k_1^*, \ldots, k_M^*\}$ and trained model.

---

IDs. Dataset $\mathbf{D}$ is a more complex and higher-dimensional simulated dataset inspired by biological measurements of gene expression and protein abundance. It allows us to validate our method in a setting similar to real data. Details on the simulation processes are provided in A.3. Since our method and some baselines require sufficiently capable autoencoders, we performed an architecture and hyperparameter search for simulation $\mathbf{D}$. Full details on the objectives, search space, training, and final selected parameters are provided in A.4.1.

### 4.1.1 ROBUSTNESS TO HYPERPARAMETER CHOICE AND DATA CHARACTERISTICS

**Experimental setup.** We evaluated the robustness of our method's general ID estimation capability with a comprehensive hyperparameter grid search on dataset $\mathbf{A}$ over distortion budget $\lambda$, rank reduction frequency $\tau$, rank reduction threshold $\gamma$, and patience $\pi$. We further tested FiGuRO's stability across a range of data characteristics, including varying sample sizes, different generative distributions, levels and types of nonlinearity, and increasing levels of noise and dropout. Details on the experimental design and training are provided in A.7 and A.9. We further tested different distortion metrics with respect to nonlinearity and dropout, which affect data distribution and potentially the appropriateness of the metrics. Metrics are described in A.5. Lastly, we demonstrate the training dynamics on popular 3D Manifold datasets: Sphere, Swiss Roll, and S-curve.

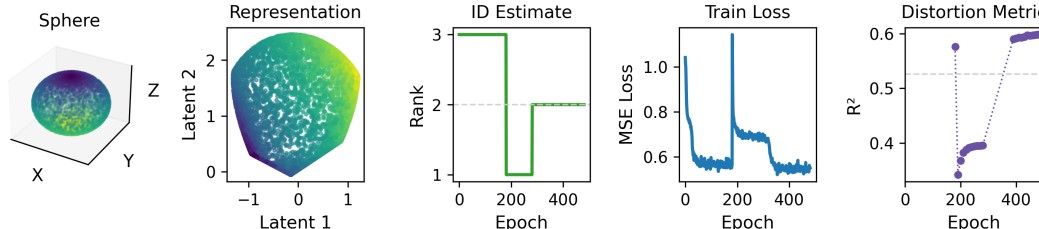

Figure 1: **FiGuRO's ID estimation over time on 3D Sphere Manifold.** From left to right: 3D sphere data space with ID 2, learned latent representation with 2 dimensions, rank over time (epochs) as the ID estimate with a dashed line on true ID 2, training mean squared error (MSE) loss over epochs, and distortion metric ($R^2$) over epochs with dashed line depicting the distortion budget (minimum possible $R^2$).

**Results.** Across the entire hyperparameter sweep (N=1080 runs), our method gave a robust average estimate of $4.89 \pm 0.01$ standard error of the mean (SEM) for a true ID of 5. Summary statistics of the sweep are provided in Supplementary Table 1. This sweep revealed a mild dependence on $\lambda$, confirming that the distortion budget is the main driver of the stopping criterion. Dependency on $\gamma$ was negligible. This stability allowed us to select a single configuration balancing performance and speed $\{\lambda = 0.05, \tau = 10, \gamma = 0.01, \pi = 10\}$ for subsequent experiments. The method was also robust to most data characteristics. Accurate estimates were achieved for sample sizes N$\geq$5000 (see A.9.2 for a discussion on general sample size requirements). Certain generative distributions (Beta, Gumbel) and high degrees of nonlinearity led to underestimation. Conversely, high levels of dropout and noise (signal-to-noise ratio $< 7$) caused the method to overestimate the true ID (Supplementary Figure 1). Dropout also had a significant effect on the robustness of different distortion metrics. Increases of ID estimates with dropout were lowest among $R^2$ and the Explained Variance Score. Given different types of nonlinearities, $R^2$, MSE, and RMSE were most robust with smallest deviations from the true ID (Supplementary Figure 2). As a result, we identified $R^2$ as the best overall distortion metric for our experiments. Applied to popular 3D Manifolds, we see the necessity of rank reduction *and* increase. Figure 1 and Supplementary Figure 3 show that ranks often dropped too low before being increased to the true ID.

### 4.1.2 ID ESTIMATION BENCHMARKS

**Experimental setup.** We benchmarked our approach's ID estimation performance on both uni-modal and multi-modal simulated datasets. For the uni-modal benchmark, we used individual modalities of $\mathbf{D}$. The two modalities $\mathbf{D_1}$ and $\mathbf{D_2}$ have the same high ambient dimension and similar ground truth IDs. However, they were designed to have distinct characteristics: $\mathbf{D_1}$ is corrupted by high levels of noise, while $\mathbf{D_2}$ is generated with a higher degree of nonlinearity (details in A.3), representing common challenges for classic ID estimation techniques. We compare our method to a range of traditional and neural network-based ID estimators. For estimators with tunable parameters, we report the range of results to show their sensitivity. Our multi-modal simulated datasets $\mathbf{B}$ provide varying IDs of the different subspaces to thoroughly test our method's capability of picking up on such relationships. We compare with a number of multi-view data decomposition methods. Implementations of baselines are explained in A.10. Architecture and training of our method can be found in A.8.

**Results.** As shown by the results in Table 2, most existing ID estimation methods are highly sensitive to specific data characteristics, yielding dramatically different estimates for the two modalities despite similar true IDs of $\mathbf{D}$. Global linear methods like PCA severely overestimated the dimension of the noisy $\mathbf{D_1}$ modality. Conversely, most local methods systematically underestimated the ID of the non-linear $\mathbf{D_2}$. Other neural network approaches, such as Rank Reduction Autoencoders (Mounayer et al., 2025), proved unstable, with estimates varying strongly depending on the choice of the rank reduction threshold $\gamma$. In contrast, our method provided good estimates for $\mathbf{D_1}$ and slightly underestimated ranks for $\mathbf{D_2}$, overall the most stable estimates closest to ground truth. On simulation $\mathbf{D}$, our method again successfully recovered IDs in the correct ballpark and with small

variance. For a true split of $6 - 5 - 6$ (shared - private 1 - private 2), our approach predicts IDs of $4.3 \pm 0.3 - 8.0 \pm 0.6$ (shared), $4.3 \pm 0.3 - 6.0 \pm 0.6$ (private 1), and $4.3 \pm 0.3 - 6.0 \pm 0.6$ (private 2) for three random seeds and distortion budget $\lambda$, ranging from 0.1 to 0.005. In the multi-modal case, most baselines show a high average deviation from the ground truth (GT) over the four datasets (Table 2). FiGuRO demonstrates a strong ability to recover the underlying dimensional structure of multi-modal data. Figure 2A shows that across all four subsets of $\mathbf{B}$, the estimated ranks for the shared and specific subspaces largely preserve the scales and relationships between the ground truth dimensions (Table 1), albeit sometimes underestimating the true ID ($\mathbf{B}_{i1}$ shared and $\mathbf{B}_l$ private subspace 1).

Table 2: **Comparison of estimated ranks for joint and individual components across multi-modal decomposition methods for simulated datasets B.** GT stands for ground truth ID for each generative subspace, defined by type Joint, P1 (private 1), and P2 (private 2). The remaining columns present baseline methods from multi-view data decomposition and our own method at the end. Best estimates (closest to GT) are highlighted in bold, second best underlined. We report average deviation from GT at the end including the standard error of the mean SEM. For our method, we used a distortion budget of $\lambda = 0.05$.

| Data | GT | Type | JIVE | AJIVE | SLIDE | ShIndICA | Ours |
|------|-----|------|------|-------|-------|----------|------|
| $\mathbf{B}_s$ | 2 | Joint | **1** | **3** | **1** | **1** | 3.6 $\pm$ 0.6 |
| | 3 | P1 | 11 | 197 | 10 | 10 | **1.0 $\pm$ 0.0** |
| | 5 | P2 | 9 | **6** | 8 | 8 | **6.2 $\pm$ 1.8** |
| $\mathbf{B}_{i1}$ | 20 | Joint | 1 | **21** | 1 | 1 | 13.4 $\pm$ 0.4 |
| | 2 | P1 | 23 | **2** | 22 | 22 | 4.4 $\pm$ 0.2 |
| | 2 | P2 | 23 | **3** | 22 | 22 | 4.0 $\pm$ 0.5 |
| $\mathbf{B}_{i2}$ | 2 | Joint | **1** | **3** | **1** | **1** | 7.0 $\pm$ 0.0 |
| | 2 | P1 | 12 | 196 | 11 | 11 | **1.0 $\pm$ 0.0** |
| | 20 | P2 | 23 | **20** | 22 | 22 | 19.4 $\pm$ 1.5 |
| $\mathbf{B}_l$ | 20 | Joint | 1 | 21 | 1 | 10 | **19.2 $\pm$ 0.7** |
| | 20 | P1 | 41 | 40 | 40 | 31 | **12.8 $\pm$ 1.0** |
| | 20 | P2 | 41 | 42 | 40 | 31 | **15.2 $\pm$ 1.3** |
| Avg deviation from GT | | | 12.4 | 36.3 | 11.8 | 9.5 | **2.9** |
| SEM | | | 2.5 | 21.7 | 2.4 | 2.1 | **0.7** |

### 4.1.3 SUBSPACE DISENTANGLEMENT

**Experimental setup.** If the ID estimation of subspaces is successful, we also assume that information will be correctly separated into shared and modality-specific subspaces. We evaluated this disentanglement on our simulations $\mathbf{B}$ and again compared the results to multi-view data decomposition baselines. We also performed an ablation study to determine the impact of Algorithm 1 compared to multi-modal rank reduction. Furthermore, we tested whether additional loss terms (Frobenius norm, L1, L2) could increase disentanglement by enforcing orthogonality.

**Results.** Figure 2B describes how well information of each set of generative variables of $\mathbf{B}$ can be predicted from FiGuRO's subspaces. For the labels from shared space and modality 2, we see very high predictability from the corresponding subspace with much lower values from the others. While this difference is not as high for label 1, modality-specific subspace 1 still gives the consistently highest predictability. This is not the case for the data decomposition baselines. Supplementary Figures 4-7 show that joint and individual information is consistently swapped. This suggests a classic failure mode of methods assuming the strongest signal to lie in the joint space. Our ablation study (Supplementary Table 3) showed that a naive "rank-reduction-only" approach universally collapses in multi-modal settings, demonstrating the importance of our proposed algorithm. Adding loss terms such as the Frobenius norm for orthogonality and an L1 regularization provided no reliable trends in rank estimation, accuracy, or predictability across our multi-modal simulations, rather decreasing performance (Supplementary Tables 4-7). However, adding L2 regularization to the first layer of the decoder helped enforce disentanglement in edge cases where modalities were truly independent with no shared information (Supplementary Table 8).

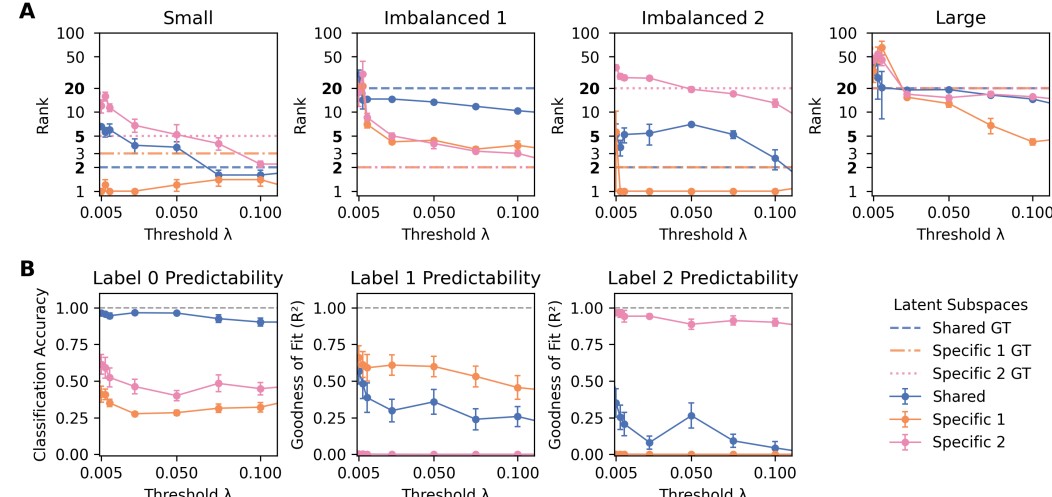

Figure 2: **Multi-modal ID estimation and disentanglement of 4 datasets B with varying true IDs.** The x axis presents the distortion threshold $\lambda$. **(A)** Log-scale estimated ranks (mean $\pm SEM$, $N = 5$ seeds) for shared (blue) and modality-specific (orange, pink) subspaces. Ground truth (GT) IDs are depicted as dashed lines. Values closer to the GT lines of the same color indicate better performance. **(B)** Average predictability of shared and private information over all random seeds and datasets ($N = 20$). The class of the shared space (label 0) is evaluated as classification accuracy. Labels 1 and 2 represent the mean value of modality-specific generative hidden vectors. Their predictability is evaluated with the $R^2$. Metrics are defined in A.5.

## 4.2 APPLICATION TO REAL DATA

Finally, to demonstrate FiGuRO's utility on real-world data, we applied it to several multi-modal datasets with diverse modalities. We performed qualitative analyses evaluating whether the information separation and dimensions inferred by FiGuRO matched reported findings. We also performed quantitative analyses through downstream task prediction.

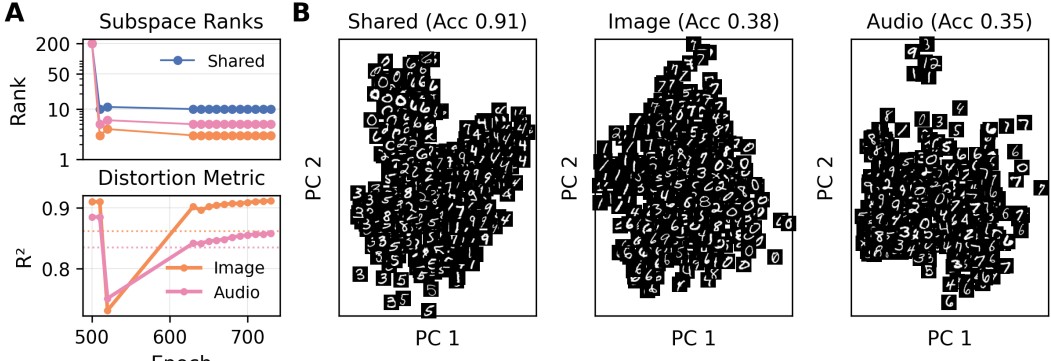

Figure 3: **Disentanglement of semantic content from modality-specific style on Audio MNIST (A)** Rank estimation and distortion metric (goodness of fit $R^2$) over epochs. Colors indicate the subspaces in the rank plot and modalities in the distortion plot. Dashed lines indicate the distortion threshold. **(B)** Latent representations per subspace. We show the first two components of PCAs, with MNIST images instead of dots. Titles indicate the subspace and show classification accuracy of digits.

### 4.2.1 QUALITATIVE EVALUATION ON $> 2$ MODALITIES

**Experimental setup.** The Non-Invasive Multimodal Foetal ECG-Doppler (NInFEA) dataset (Sulas et al., 2021) contains 60 synchronized recordings of maternal and fetal physiology from high-density electrocardiography (ECG) from the maternal abdomen and thorax, a maternal respiration signal, and Pulsed-Wave Doppler (PWD) ultrasound of the fetal heart. While we do not have ground truth IDs for this data, we can evaluate the relative amounts of shared information between modalities relying on clinical intuition. We applied FiGuRO on all pairs of the four modalities and report the fraction of shared over total ranks. We also trained a model on three modalities at once. Architecture and training are adjusted to the temporal and image modalities (see A.8).

**Results.** Applying FiGuRO to the NInFEA dataset allowed us to quantitatively assess information overlap between modalities. Our estimated shared-to-total rank ratios (Supplementary Table 9) confirm established clinical and physiological knowledge: the fetal ECG (fECG) and Pulse-Wave Doppler (fPWD) show the largest fraction of shared information ($0.54$), reflecting their common underlying fetal cardiac process (Sulas et al., 2021). Other pairs, such as the maternal ECG (mECG) and respiration (mR), also showed the expected synchronization overlap ($0.47$) (Yasuma & Hayano, 2004). Conversely, the low overlap between fPWD and the maternal modalities (mECG/mR) was also consistent with expectations. Results from a three-modality version (fECG, mR, and fPWD) again highlighted the strongest overlap between fECG and fPWD (Supplementary Table 10), suggesting the feasibility of applying FiGuRO to settings with more than two modalities.

### 4.2.2 PERFORMANCE ON DOWNSTREAM TASKS

**Experimental setup.** We applied FiGuRO as well as the multi-view decomposition baselines to Audio MNIST (Becker et al., 2023) and trained uni-modal autoencoders. See the Appendix for details on data preprocessing (A.11), architectures (A.4), and training (A.8). We report ID estimates and evaluated disentanglement and downstream task performance as the classification accuracy of the provided labels.

Table 3: **Multi-modal downstream task performance evaluation (classification).** We report classification accuracies (Appendix A.6.1) of the main prediction task per subspace, i.e. $Acc_s$ being the accuracy on the shared subspace. Arrows indicate whether a higher or lower accuracy is better, based on the assumption that the predicted classes are expected to be part of the shared information. $i$: image modality, $a$: audio. For any predictions from models we trained, we report the mean $\pm$ SEM. For our method, we used a distortion budget of $\lambda = 0.05$. Random guessing would give a base accuracy of 0.1 for Audio MNIST.

| Dataset | Metric | Unimodal | JIVE | AJIVE | SLIDE | ShIndICA | **Ours** |
|---------|--------|----------|------|-------|-------|----------|----------|
| Audio | $Acc_s \uparrow$ | – | 0.42 | **0.99** | 0.28 | 0.23 | $0.94 \pm 0.01$ |
| MNIST | $Acc_1 \downarrow$ | $0.66 \pm 0.04$ | 0.63 | 0.41 | 0.77 | 0.46 | $0.49 \pm 0.06$ |
| $(i, a)$ | $Acc_2 \downarrow$ | $0.65 \pm 0.03$ | 0.81 | 0.65 | 0.72 | 0.71 | $0.44 \pm 0.02$ |
| | $Acc_{all} \uparrow$ | $0.80 \pm 0.04$ | 0.64 | 0.99 | 0.45 | 0.36 | $0.97 \pm 0.01$ |

**Results.** While we do not know the ground truth ID of real datasets, we have a good intuition of what to expect for Audio MNIST. In the literature, MNIST is assumed to have an ID of $7 - 25$ (Pope et al., 2020). Supplementary Table 11 shows that baselines are far off from this expectation, but FiGuRO is within the range with an average rank ($k_s + k_{image}$) of $11.4 \pm 0.6$. Most of the baselines (except for AJIVE) also perform badly on the label classification, again swapping joint and individual information. Our proposed method correctly learns digit classification on Audio MNIST in the shared subspace only (see Figure 3), and consistently outperforms uni-modal embeddings (Supplementary Table 3). Looking at reconstruction performance on Audio MNIST, test performance is very high, with losses only increasing by $0.2\%$ even though the latent space has decreased 36-fold (Supplementary Table 12, Supplementary Figures 9, 10).

## 5 CONCLUSION

In this work, we addressed the fundamental challenge of estimating the intrinsic dimensions (IDs) of multi-modal data. We introduced Fidelity-guided Rank Optimization (FiGuRO), a framework that explicitly and dynamically learns the dimensionalities of shared and private subspaces in mult-imodal data. We demonstrated empirically that FiGuRO is robust to hyperparameters and differ-ent data characteristics, and significantly outperforms traditional uni-modal estimators on complex, high-dimensional simulated data. In the multi-modal setting, FiGuRO successfully recovered the relative scales and ratios of shared and private IDs and disentangled modality-specific information in scenarios where classical data decomposition methods failed. Our findings are further validated on real-world datasets with diverse modalities, including time series, images, and audio. Notably, we demonstrated on Audio MNIST that FiGuRO can learn low-dimensional decompositions success-fully separating shared and specific information without significant performance loss.The efficacy of our framework, like other neural network approaches, relies on an underlying autoencoder architec-ture that is sufficiently expressive to model the data. This requires an adequate number of training samples and inherits the general challenges of training and balancing multiple modalities, partic-ularly when they exhibit different signal-to-noise ratios or loss scales. While we did not observe any failures with distortion metrics tested in our experiments, FiGuRO could fail when distortion metrics are not chosen appropriately for the data at hand. We also note that the inherent randomness of neural networks brings along some uncertainty to the ID estimation. For robust estimation, we recommend using FiGuRO to estimate bounds under low and high distortion budgets.

Altogether, we believe that our method provides a critical missing piece to multi-modal learning: a principled method to quantify informational complexity, thereby moving the field from static de-sign toward the deliberate construction of more interpretable models. Furthermore, low-rank bot-tlenecks provide meaningful regularization and may prove useful for generalization and improved interpretability, which are crucial for machine learning in biology and medicine. Looking forward, the dynamic nature of our framework also makes it uniquely suited for continual learning scenarios. FiGuRO can be extended to settings where models must adapt their latent capacity online in response to new data from increasingly heterogeneous sources, without the need for complete retraining.

### REPRODUCIBILITY

We are committed to make our work reproducible. The complete source code is provided as sup-plementary materials. Our method FiGuRO is formally described in Section 3. The theoretical underpinnings including assumptions and proofs, are provided in the Appendix. All experimental setups, including data simulations, model architectures, training hyperparameters, and implemen-tation details for each dataset and baseline method, are thoroughly documented in the Appendix. Details regarding the preprocessing of real-world datasets (NInFEA, Audio MNIST) are also pro-vided in the Appendix.

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

# A APPENDIX

## A.1 THEORETICAL GUARANTEES FOR ID ESTIMATION

Our empirical results are supported by a theoretical framework grounded in the Manifold Hypothesis and Rate-Distortion Theory. We formalize this by first stating our core assumptions, followed by theorems that provide guarantees for the convergence, correctness, and sample complexity of FiGuRO. Our theoretical results hold under the following assumptions:

**Assumption 1** (Data Generation). Let the "clean" data $\mathbf{Y}$ be sampled from a distribution $p(x)$ with support on an $r$-dimensional compact manifold $\mathcal{M} \subset \mathbb{R}^n$. Let the observed data be corrupted by i.i.d. additive noise

$$X = Y + \epsilon$$

where the noise $\epsilon$ is drawn from a distribution with zero mean ($\mathbb{E}[\epsilon] = 0$) and a covariance matrix $\Sigma_\epsilon = \sigma^2 I_D$, with $\sigma^2$ representing the noise variance. The variance of the noise is smaller than the geometric variance of the Manifold.

**Assumption 2.** For any multi-modal data, we further assume that the number of generative variables is greater than 0, such that the modalities are not fully independent.

**Assumption 3** (Sufficient Capacity). Given Assumption 1, we assume our autoencoder architecture, defined by the encoder family $\mathcal{F}$ (parameterized by $\phi$) and decoder family $\mathcal{G}$ (parameterized by $\theta$), constitutes a **sufficient function approximator**. This holds if, for any arbitrarily small precision $\epsilon > 0$, there exists a set of parameters $(\phi, \theta)$ and a latent dimension $l \geq r$ such that the expected reconstruction distortion $\mathbb{E}[d(X, \hat{X})]$ can be made arbitrarily close to the irreducible noise floor:

$$\mathbb{E}_{p(x)}\left[d(x, g_\theta(f_\phi(x)))\right] \leq \sigma^2 + \epsilon$$

**Assumption 4** (Fidelity Proxy). We assume the coefficient of determination, $R^2$, or any other metric chosen, is a suitable and stable proxy for reconstruction fidelity. Specifically, for a given distortion budget $\lambda > 0$, the condition $R^2(X, \hat{X}) \geq R_0^2 - \lambda$ implies that the true reconstruction distortion $\mathbb{E}[d(X, \hat{X})]$ on the manifold $\mathcal{M}$ is bounded.

**Assumption 5** (Compactness). We assume that the $r$-dimensional manifold $\mathcal{M}$ is compact and its generative variables are $i.i.d.$. Compactness implies that there is no sparsity among the generative variables of the data.

**Theorem 1.** Under Assumptions 3 and 4, the rank optimization phase of the FiGuRO algorithm is **guaranteed to converge** in a finite number of epochs, returning a final set of ranks $\{k_s^*, k_1^*, ..., k_m^*\}$.

*Proof.* The rank of any adaptable layer is a positive integer bounded between 1 and its initial maximum rank. These bounds are updated and tightened during training. The algorithm only changes ranks if the fidelity condition is met or violated. The patience parameter $\pi$ ensures that if the ranks remain unchanged for $\pi$ consecutive checks, the optimization process terminates. As a result, ranks cannot oscillate indefinitely and the algorithm is guaranteed to halt. $\square$

**Theorem 2.** Let the data be generated from a distribution supported on a dense $r$-dimensional manifold $\mathcal{M}$ (Assumption 5). Under Assumptions 3 and 4, and given a sufficient number of samples $N$ (as defined in Corollary 3), there exists a sufficiently small distortion budget $\lambda$ such that the rank $k^*$ returned by FiGuRO satisfies $\mathbb{E}[k^*] \approx r$.

*Proof.* The proof relies on the connection to Rate-Distortion theory. If the estimated rank $k^*$ is less than the true ID $r$, the autoencoder is forced to discard information essential for describing the manifold, causing the reconstruction distortion to violate the budget $\lambda$. Conversely, if $k^* > r$, the model is still over-parameterized. The rank reduction mechanism can then prune redundant dimensions without violating the fidelity budget. The algorithm thus converges to a rank that balances model capacity (rate) with reconstruction quality (distortion), which corresponds to the intrinsic dimension of the manifold. This aligns with recent work showing that globally optimized autoencoders can provably learn the correct manifold dimension (Zheng et al., 2023). $\square$

**Corollary 3.** For the guarantee in Theorem 2 to hold, the number of training samples $N$ must satisfy the sample-to-capacity ratio:

$$N \geq \frac{\alpha \cdot \mathcal{C}_e}{k^*} \text{ with } \alpha \geq 10$$

where $k^*$ is the estimated ID, $\mathcal{C}_e$ is the number of parameters in the encoder, and $\alpha$ is a constant factor termed the load (Schuster & Krogh, 2021).

*Proof.* If $N$ is insufficient, the autoencoder will overfit to the training data. This leads to an unreliable fidelity signal ($R^2$), as the model's performance on the validation set (which guides the fidelity check) will diverge from its performance on the training set. A stable and generalizable fidelity signal is necessary for the rank optimization mechanism to correctly probe the data's true dimensionality. □

**Theorem 4.** Let $\mathcal{M}$ be a $d$-dimensional compact Manifold embedded in an ambient space $\mathbb{R}^n$. Under Assumption 1 the top $k^*$ ranks represent the geometric variance of the Manifold and discard noise.

*Proof.* The principal components of the data are a combination of geometric and noise variance. As long as the noise level $sigma$ is not excessively high relative to the manifold's curvature, the singular values associated with the manifold's structure are expected to be significantly larger than those associated purely with noise. Then, by penalizing reconstruction error via the distortion metric, the network is incentivized to first capture the geometric variance. The rank reduction process then acts as a form of denoising, progressively discarding the smaller singular values that correspond to the noise subspace. □

### A.2 THEORETICAL GUARANTEES FOR DISENTANGLEMENT

While our primary objective $\mathcal{L}_{\text{recon}}$ (Eq. 5 in main paper) does not include an explicit term for minimizing mutual information between subspaces, we posit that disentanglement emerges as an optimal solution from the interplay between our model architecture (Eq. 4 in main paper) and the rate minimization objective of FiGuRO (Algorithm 1). We formalize this using the principles of Rate-Distortion Theory and Information Bottlenecks (IB).

**Assumption 6.** We assume that the rank $k^*$ of a latent subspace, as estimated by FiGuRO, serves as a proxy for its information content, or *Rate R*. This is an extension of the Minimum Description Length (MDL) principle, where the rank $k^*$ is the minimal number of parameters (degrees of freedom) required to describe the data, such that $k^* \propto R \approx I(Z; X)$. Given this, the FiGuRO algorithm can be interpreted as solving the following optimization problem for latent representations $H_s, H_1, H_2$ derived from inputs $X_1, X_2$:

1. **Minimize Total Rate (Rank):** $\min_{k_s, k_1, k_2} R_{\text{total}} \approx k_s + k_1 + k_2$

2. **Subject to Distortion Constraint (Fidelity):** The reconstruction error for each modality must remain bounded by the distortion budget $\lambda$, which implies sufficient information preservation:

$$I(H_s, H_1; X_1) \geq H(X_1) - \mathcal{D}_1$$
$$I(H_s, H_2; X_2) \geq H(X_2) - \mathcal{D}_2$$

where $H(X)$ is the entropy (total information) of a modality and $\mathcal{D}$ is the information loss permitted by $\lambda$.

We further define the "true" information components of the data (assuming no synergistic information):

- **Shared Information:** $S = I(X_1; X_2)$
- **Private Information (Modality 1):** $P_1 = H(X_1|X_2) = H(X_1) - S$
- **Private Information (Modality 2):** $P_2 = H(X_2|X_1) = H(X_2) - S$

The total information required to reconstruct both modalities is $H(X_1, X_2) = S + P_1 + P_2$.

**Theorem 5** (Optimality of the Disentangled Solution). Under the FiGuRO optimization objective (minimizing total rank s.t. reconstruction fidelity) and given the architectural constraints (Eq. 4), the unique optimal (lowest-rank) solution that satisfies the reconstruction constraint is the disentangled solution, where:

$$I(H_s) \rightarrow S$$
$$I(H_1) \rightarrow P_1$$
$$I(H_2) \rightarrow P_2$$

This implies $I(H_1; H_s) \rightarrow 0$, $I(H_2; H_s) \rightarrow 0$, and $I(H_1; H_2) \rightarrow 0$.

*Proof.* The total information $S + P_1 + P_2$ must be captured by the latent subspaces $H_s, H_1, H_2$ to satisfy the reconstruction fidelity constraint. The total rate (rank) to be minimized is $R_{\text{total}} \approx k_s + k_1 + k_2 \approx I(H_s) + I(H_1) + I(H_2)$. We compare two possible solutions that both satisfy reconstruction: $\square$

1. **Solution 1: The Disentangled (Optimal) Allocation**
   - $H_s$ is allocated the shared information: $I(H_s) \approx S$.
   - $H_1$ is allocated the private information for $X_1$: $I(H_1) \approx P_1$.
   - $H_2$ is allocated the private information for $X_2$: $I(H_2) \approx P_2$.
   - **Check Constraints:**
     - Decoder $g_1$ receives $H_s \oplus H_1$, which contain $(S, P_1)$. This is sufficient to reconstruct $X_1$ (as $H(X_1) = S + P_1$).
     - Decoder $g_2$ receives $H_s \oplus H_2$, which contain $(S, P_2)$. This is sufficient to reconstruct $X_2$ (as $H(X_2) = S + P_2$).
   - **Total Rate:** $R_{\text{optimal}} \approx S + P_1 + P_2$.

2. **Solution 2: A Non-Disentangled (Suboptimal) Allocation** Consider a solution where the model relies only on the private pathways and $H_s$ is pruned (i.e., $k_s \rightarrow 0$).
   - $H_s$ is allocated no information: $I(H_s) = 0$.
   - To reconstruct $X_1$, $H_1$ *must* now capture all of $X_1$'s information: $I(H_1) \approx H(X_1) = S + P_1$.
   - To reconstruct $X_2$, $H_2$ *must* now capture all of $X_2$'s information: $I(H_2) \approx H(X_2) = S + P_2$.
   - **Check Constraints:**
     - Decoder $g_1$ receives $(0, S + P_1) \rightarrow$ Reconstructs $X_1$. (Satisfied)
     - Decoder $g_2$ receives $(0, S + P_2) \rightarrow$ Reconstructs $X_2$. (Satisfied)
   - **Total Rate:** $R_{\text{suboptimal}} \approx 0 + (S + P_1) + (S + P_2) = (S + P_1 + P_2) + S$.

**Conclusion:** The non-disentangled solution (Solution 2) has a total rate of $R_{\text{suboptimal}} \approx R_{\text{optimal}} + S$. Its total rank $k_1 + k_2$ is strictly greater than the total rank $k_s + k_1 + k_2$ of the disentangled solution (Solution 1) by approximately $\text{ID}(S)$, the intrinsic dimension of the shared information.

Because the FiGuRO algorithm (Algorithm 1) is an optimization process designed to find the minimal total rank ($R_{\text{total}}$) that satisfies the fidelity constraint, it will actively penalize any redundant, non-disentangled solution (like Solution 2). The optimization will therefore converge to the most efficient (minimal-rank) representation, which is the disentangled solution (Solution 1) where information is allocated non-redundantly.

### A.3 DATA SIMULATION

#### A.3.1 UNI-MODAL PARAMETRIC $\mathbf{A}$

We created a simple but highly flexible simulation for generating a single data modality controlled by data characteristics as parameters. This allows for rapid testing of model performance across a wide range of data structures and complexities. The simulation generates an observed data matrix $\mathbf{A} \in \mathbb{R}^{N \times n}$ from a latent representation matrix $\mathbf{Z} \in \mathbb{R}^{N \times k}$, where $N$ represents the number of samples, $n$ the ambient dimension, and $k$ the latent dimension (the ID). We first sample $\mathbf{Z}$ from a chosen distribution $P$ with options {Beta, Gaussian, Poisson, Binomial, Gumbel, Uniform, Weibull}. The latent variables undergo no, one, or more rounds (controlled by the nonlinearity level) of nonlinear transformations. The specific function is determined by the nonlinearity type parameter $f(x) \in \{x^2, \max(0, x), \frac{1}{1+e^{-x}}, \sin(x)\}$. The transformed latent variables $\mathbf{z}'$ are linearly projected into the higher-dimensional data space using a sparse weight matrix $\mathbf{W} \in R^{k \times n}$ controlled by a connectivity parameter. Finally we add noise $\epsilon$ with a desired variance and apply dropout.

$$\mathbf{A} = \mathbf{z}'\mathbf{W} + \epsilon \text{ with } \mathbf{z}' = f(\mathbf{z}), \mathbf{z} \sim P, \epsilon \sim \mathcal{N}(0, \sigma) \tag{1}$$

#### A.3.2 MULTI-MODAL PARAMETRIC $\mathbf{B}$

To evaluate our model's ability to handle multi-modal data, we extend the uni-modal framework to generate two data matrices, $\mathbf{B}_1$ and $\mathbf{B}_2$. The key difference is the use of a composite latent structure comprising both shared variables $\mathbf{Z}_s \in \mathbb{R}^{N \times k_s}$ that influence both modalities and modality-specific variables $\mathbf{Z}_m \in \mathbb{R}^{N \times k_m}$ that affect only one. Based on our experimental configurations, the shared variables $\mathbf{Z}_s$ are sampled from either a Binomial distribution (when $k_s = 2$) or a Gaussian Mixture Model for $k_s > 2$. The modality-specific variables $\mathbf{Z}_{m1}$ and $\mathbf{Z}_{m2}$ are sampled from Poisson and Weibull distributions, respectively. For each modality m, a complete latent representation is formed by concatenating the shared and modality-specific variables. This combined representation is then projected into the data space using a modality-specific sparse weight matrix $\mathbf{W}_m$. For simplicity, the nonlinear transformation step was omitted in these experiments. As in the uni-modal case, each data matrix is independently corrupted by adding Gaussian noise and applying a dropout mask.

#### A.3.3 MULTI-OMICS $\mathbf{D}$

To generate a realistic multi-modal dataset with a known causal structure, we designed a multi-stage simulation inspired by a limited set of known processes in gene expression and translation. The result is the multi-omics data simulation $\mathbf{D}$. The full procedure and parameterization are detailed in Algorithm 2.

**Genomic and cellular architecture:** First, we establish a fixed genomic architecture where 20,000 genes are organized into co-regulated gene clusters and programs, including a core set of housekeeping genes. We then simulate a cell lineage tree originating from three distinct stem cell lines. Cellular differentiation is modeled as the progressive and stochastic silencing of gene programs, which defines each cell type's identity and its baseline gene expression potential.

**Simulation flow:** With the genomic architecture fixed, we simulate the molecular state for each cell in a causal cascade. We determine a ground-truth chromatin accessibility profile for each cell. This profile is then perturbed by cell-specific variables simulating cell cycle phase and a general stress response. The resulting accessibility profile, along with gene-specific transcription efficiencies and cell-specific DNA damage, dictates the "true" mRNA counts. In turn, these mRNA counts determine protein abundance via translational efficiencies (dependent on amino acid composition and ribosome availability) and protein degradation rates. Finally, each modality is subjected to a separate technical noise model that simulates artifacts like capture efficiency, batch effects, and stochastic dropout to produce the final observed data matrices.

---

**Algorithm 2** Multi-Omics Data Simulation $\mathbf{D}$

---

1: **Inputs:** $N_{genes}$, $N_{cells}$, $N_{stemcells}$
2: **Outputs:** Modality matrices $\mathbf{D_1}$, $\mathbf{D_2}$, and causal variables $\mathbf{Z}$.

3: $\qquad\qquad\qquad\qquad\qquad$ ▷ **Part 1: Generate Fixed Genomic and Cell Type Architecture (seed=0)**
4: **Define Gene Properties:**
5: $\quad$ For each gene $g \in \{1, ..., N_{genes}\}$:
6: $\qquad$ Gene length $L_g \sim \lfloor (\text{NegativeBinomial}(300, 0.02))^{3.9} \rfloor + 150$
7: $\qquad$ Base expression $\mu_{base,g} \sim \text{NegativeBinomial}(1000, 0.01) + 1$
8: $\qquad$ Transcription probability $\tau_{trans,g} = f_{trans}(L_g) + \mathcal{N}(0, 0.05^2)$
9: $\qquad$ mRNA degradation prob. $\delta_{RNA,g} = f_{deg,RNA}(L_g) + \mathcal{N}(0, 0.05^2)$
10: $\qquad$ Protein length $L'_g = \lfloor L_g/3 \rfloor$
11: $\qquad$ Translation ease $\phi_{AA,g} = f_{ease}(\text{AA composition of } g)$
12: $\qquad$ Translation probability $\tau_{prot,g} = f_{trans}(L'_g)$
13: $\qquad$ Protein degradation prob. $\delta_{PROT,g} = f_{deg,PROT}(L'_g)$
14: **Define Regulatory Structure:**
15: $\quad$ Generate gene cluster assignment matrix $\mathbf{M}_{C \to G}$
16: $\quad$ Generate gene program matrix $\mathbf{M}_{P \to C}$ where $M_{P \to C}[p, c] \sim \text{Bernoulli}(0.1)$
17: $\quad$ Generate cell type hierarchy matrix $\mathbf{M}_{CT \to P}$ via iterative program silencing
18: $\quad$ Compute cell type to gene activity: $\mathbf{M}_{CT \to G} = \text{threshold}(\mathbf{M}_{CT \to P} \mathbf{M}_{P \to C} \mathbf{M}_{C \to G})$
19: **Define Perturbation Effects:**
20: $\quad$ Stress closure vector (by gene) $\mathbf{c}_{stress} = \text{cluster\_closure} \cdot \mathbf{M}_{C \to G}$
21: $\quad$ Cell cycle amplification matrix $\mathbf{E}_{cc}$ and openness matrix $\mathbf{M}_{open,cc}$

22: $\qquad\qquad\qquad\qquad\qquad\qquad\qquad\qquad\qquad\qquad\qquad\qquad$ ▷ **Part 2: Simulate All Cells**
23: **Sample Cell-Specific Latent Variables** for $i \in \{1, ..., N_{cells}\}$:
24: $\quad$ Cell type $ct_i \sim \text{Categorical}(\mathbf{1}/N_{CT})$
25: $\quad$ Stress level $s_i \sim \text{Bernoulli}(0.05)$
26: $\quad$ Cell cycle phase $cc_i \sim \text{Categorical}([0.1, 0.2, 0.3, 0.4])$
27: $\quad$ Transcription activity $a_{trans,i} \sim \text{Clamp}(\text{Poisson}(4), 0, 9) + 1$
28:
29: $\quad$ Baseline open chromatin: $\mathbf{O}_{base,i,:} = a_{trans,i} \cdot \mathbf{M}_{CT \to G}[ct_i, :]$
30: $\quad$ Cell cycle modulation: $\mathbf{O}_{cc,i,:} = (\mathbf{O}_{base,i,:} + \mathbf{E}_{cc}[cc_i, :]) \odot \mathbf{M}_{open,cc}[cc_i, :]$
31: $\quad$ Final ground truth open chromatin: $\mathbf{O}_{i,:} = \mathbf{O}_{cc,i,:} \odot (1 - s_i \cdot \mathbf{c}_{stress})$
32:
33: $\qquad\qquad\qquad\qquad\qquad\qquad\qquad\qquad\qquad\qquad\qquad\qquad$ ▷ **Transcription (RNA-seq)**
34: $\quad$ DNA damage $p_{dmg,i} \sim \text{Beta}(1, 2) \cdot \text{level}(ct_i)/10$
35: $\quad$ Potential transcription: $\mathbf{R}_{pot} = \mathbf{O} \odot \boldsymbol{\mu}_{base}$
36: $\quad$ Real mRNA counts: $\mathbf{R}_{RNA} = \lfloor \mathbf{R}_{pot} \odot (1 - \mathbf{p}_{dmg}) \odot \boldsymbol{\tau}_{trans} \odot e^{-\boldsymbol{\delta}_{RNA}} \rfloor$
37: $\quad$ Technical group assignment: $b_{RNA,i} \sim \text{Categorical}(\mathbf{1}/3)$
38: $\quad$ Observed mRNA: $\mathbf{D_{RNA,i}} = \lfloor \mathbf{R}_{RNA,i} \cdot \epsilon_{RNA}(b_{RNA,i}) \rfloor \odot (1 - d_{RNA}(b_{RNA,i}))$
39:
40: $\qquad\qquad\qquad\qquad\qquad\qquad\qquad\qquad\qquad\qquad\qquad\qquad$ ▷ **Translation (Proteomics)**
41: $\quad$ Sample protein machinery variables for cell $i$:
42: $\qquad$ Ribosome availability $f_{ribo,i} \propto \sum_{g \in \text{rDNA}} R_{RNA,i,g}$
43: $\qquad$ tRNA availability $a_{tRNA,i} \sim \text{Beta}(2, 1)$
44: $\qquad$ Proteasome activity $a_{prot,i} \sim \text{Beta}(1, 2)$
45: $\quad$ Ribosome efficiency: $\boldsymbol{\eta}_{ribo,i} = (\boldsymbol{\phi}_{AA}/0.05 \odot \boldsymbol{\tau}_{prot}) \cdot (f_{ribo,i}/\sum_g R_{RNA,i,g} \cdot a_{tRNA,i})$
46: $\quad$ Proteins translated: $\mathbf{P}_{trans} = \mathbf{R}_{RNA} \odot \boldsymbol{\eta}_{ribo}$
47: $\quad$ Real protein counts: $\mathbf{P}_{real} = \mathbf{P}_{trans} \odot e^{-(\boldsymbol{\delta}_{PROT} \odot \mathbf{a}_{prot})}$
48: $\quad$ Technical group assignment: $b_{PROT,i} \sim \text{Categorical}(\mathbf{1}/2)$
49: $\quad$ Observed protein: $\mathbf{D_{Protein,i}} = \lfloor \mathbf{P}_{real,i} \cdot \epsilon_{PROT}(b_{PROT,i}) \rfloor \odot (1 - d_{PROT}(b_{PROT,i}))$
50:
51: **Store** final matrices $\mathbf{D_1}$, $\mathbf{D_2}$ and all latent variables $\mathbf{Z}$.

---

### A.4 ARCHITECTURES

All autoencoders trained in this work consist of symmetric encoder-decoder architectures.

#### A.4.1 BASE AUTOENCODER HYPERPARAMETER SEARCH

To ensure our method's performance was not confounded by a suboptimal base model, we conducted a hyperparameter search for the autoencoder architecture and training parameters using the Optuna (Akiba et al., 2019) framework. The goal was to identify a sufficiently deep and wide architecture capable of achieving high reconstruction fidelity (i.e., near-lossless compression) when using a large, fixed bottleneck dimension. The search was performed on the RNA modality of dataset $\mathbf{D}$ (N=30,000). We ran a multi-objective optimization over 100 trials, aiming to simultaneously maximize the reconstruction goodness-of-fit (mean $R^2$) and minimize the autoencoder's validation loss (MSE). The $R^2$ is explained in A.5. The search space for the optimization included key architectural and training parameters: network depth ($\{2, 3, 4, 6\}$), width as a fraction of input dimension ($\{0.25, 0.5, 0.75, 1.0\}$), learning rate (log-uniform between $10^{-5}$ and $10^{-3}$), batch size ($\{64, 128, 256, 512\}$), weight decay (log-uniform between $10^{-6}$ and $10^{-4}$), dropout ($\{0.0, 0.1, 0.2\}$), and early stopping patience ($\{10, 50\}$). From the resulting Pareto front of optimal solutions, we selected a final configuration that offered the best balance between the two objectives. This balanced solution was identified by normalizing both objective scores across the Pareto front and selecting the trial with the highest geometric mean. The final parameters were a depth of 2, width factor 0.5, dropout 0.1, batch size 512, learning rate $10^{-5}$, weight decay $2 \times 10^{-5}$, and early stopping of 50 epochs.

#### A.4.2 UNI-MODAL SIMULATIONS

For the uni-modal parametric simulations, we employed a fully-connected autoencoder. Both the encoder and decoder consisted of 2 hidden layers. The width of these layers was set to be equal to the input data dimension (50 features), corresponding to a width factor of 1.0. A dropout rate of 0.1 was applied to all hidden layers for regularization. The central layer was our low-rank adaptable layer, initialized with a rank of 20. For the more complex and high-dimensional uni-modal omics simulation benchmark, the architecture was scaled appropriately. We used an autoencoder of the same depth (2 layers), but with a width factor of 0.5 relative to the input dimension. The adaptable layer was initialized with a highly overcomplete rank of 1000. The dropout rate was kept at 0.1.

#### A.4.3 3D SHAPES

For the 3D manifold experiments, we use the same 2-layer autoencoder with a hidden width ratio of 1.0 and dropout 0.1. The central adaptable layer was initialized with a rank of 3, matching the ambient dimension of the data.

#### A.4.4 MULTI-MODAL SIMULATIONS

For the multi-modal parametric simulations, the model consisted of modality-specific encoders feeding into three adaptable bottleneck layers representing the shared, private 1, and private 2 subspaces as described in the Methods section. Each modality's encoder and decoder had 2 hidden layers with a width factor of 1.0 relative to the input dimension (200 features) and a dropout rate of 0.1. All three adaptable layers were initialized with a rank of 100. For the higher-dimensional multi-omics simulation, a similar architecture was used but with a width factor of 0.5. To accommodate the greater complexity, the initial rank for each of the three adaptable layers was increased to 500.

#### A.4.5 NINFEA

Due to the heterogeneous data types in the NInFEA dataset, which includes both time-series (ECG, respiration) and image-based (PWD) modalities, we utilized a hybrid architecture. Each modality was first processed by a dedicated encoder pathway consisting of convolutional layers to extract relevant features. These features were then passed to a fully-connected network with 2 hidden layers, each containing 512 units, with a dropout rate of 0.1 applied for regularization. The adaptable bottleneck layers for the shared and private subspaces were all initialized with a rank of 100.

### A.4.6 AUDIO MNIST

For the Audio MNIST dataset, we processed the image modality by a convolutional encoder with two Conv2d layers (16 and 64 channels, respectively), each with a 5x5 kernel and followed by 2x2 max-pooling. This was connected to two fully-connected layers of width 800. The audio modality was similarly processed with 10x10 kernels and hidden layers of width 1600. The adaptable bottleneck layers for the shared, image-specific, and audio-specific subspaces were each initialized with a rank of 200.

### A.4.7 MM-IMDB

We used the same setup as for the high-dimensional multi-modal simulations: 2 hidden layers with a width factor of 0.5. We used and initial max ranks of 200.

## A.5 METRICS

### A.5.1 DISTORTION METRICS

**Goodness of fit $R^2$**

The coefficient of determination $R^2$ is a measure of reconstruction goodness of fit. We measure it for each feature $j$ over $10\%$ of training samples $i$ and report the average over features (observables in the data) $n$.

$$R^2 = \frac{1}{n} \sum_{j=1}^{n} \left( 1 - \frac{\sum_{i=1}^{N} (x_{ij} - \hat{x}_{ij})^2}{\sum_{i=1}^{N} (x_{ij} - \bar{x}_j)^2} \right) \tag{2}$$

**MSE and RMSE**

The Mean Squared Error (MSE) is a direct measure of distortion, calculating the average of the squared differences between the original data $x$ and the reconstructed data $\hat{x}$. It is highly sensitive to large errors due to the squaring operation. The Root Mean Squared Error (RMSE) is the square root of the MSE and is often preferred as it returns the error metric to the same scale as the original data.

$$\text{MSE} = \frac{1}{ND} \sum_{i=1}^{N} \sum_{j=1}^{D} (x_{ij} - \hat{x}_{ij})^2 \tag{3}$$

$$\text{RMSE} = \sqrt{\text{MSE}} \tag{4}$$

**Explained Variance Score**

The Explained Variance Score measures the proportion of the variance in the original data that is accounted for by the model's reconstructions. A score of 1.0 indicates that the model perfectly explains the variance of the data. It is calculated as:

$$\text{Explained Variance} = 1 - \frac{\text{Var}(x - \hat{x})}{\text{Var}(x)} \tag{5}$$

## A.6 EVALUATION METRICS

### A.6.1 CLASSIFICATION ACCURACY

We use a 5-fold cross-validation logistic regression classifier to assess how well representations linearly predict ground truth class labels. We used the sklearn implementation with max_iter=1000, class_weight='balanced', solver='liblinear', and random_state=42 to predict class labels $y_i$. The accuracy is the number of correct predictions divided by the total number of predictions.

$$Accuracy = \frac{1}{N} \sum_{i=1}^{N} \mathbb{I}(y_i = \hat{y}_i) \tag{6}$$

### A.7   FiGuRO HYPERPARAMETER SEARCH

All tested hyperparameters are reported in Table 1 along with their estimated ranks and sample sizes.

### A.8   TRAINING

In this section we describe general training frameworks and the specific hyperparameter sets used per experiment. We used the Adam optimizer and mean squared error (MSE) loss for all training runs.

#### A.8.1   UNI-MODAL SIMULATIONS

Models for the parametric simulations were trained using a learning rate of $10^{-4}$ and a weight decay of $2 \times 10^{-5}$. We used a batch size of 128 and trained for a maximum of 5000 epochs. The FiGuRO rank optimization procedure began after an initial pre-training phase with 50 epochs early stopping. For the high-dimensional omics simulations, the training configuration was adapted for the larger dataset. We used a lower learning rate of $10^{-5}$ with the same weight decay and a larger batch size of 512.

#### A.8.2   3D SHAPES

The models were trained with a learning rate of $10^{-3}$ and a weight decay of $2 \times 10^{-5}$. We used a batch size of 512 and trained for a maximum of 5000 epochs with early stopping 50.

#### A.8.3   MULTI-MODAL SIMULATIONS

The parametric simulation models were trained for a maximum of 5000 epochs (50 early stopping) with a learning rate of $10^{-4}$, a weight decay of $10^{-5}$, and a batch size of 128. For the larger multi-omics simulation, the training parameters were adjusted to a learning rate of $10^{-5}$, a weight decay of $2 \times 10^{-5}$, and a batch size of 1024.

#### A.8.4   NInFEA

The models were trained for a maximum of 5000 epochs with a batch size of 8, a learning rate of $10^{-4}$, and no weight decay. The rank adaptation process was initiated after an initial training phase, which was run with an early stopping patience of 100 epochs. Once optimization began, the final rank convergence was determined with a patience of 10 epochs.

#### A.8.5   AUDIO MNIST

The training process consisted of two distinct phases. In the first phase, we performed rank optimization for a maximum of 5000 epochs using a batch size of 512 and a learning rate of $10^{-3}$ with a linear decay schedule. The FiGuRO algorithm was initiated after an initial pre-training phase, which ran until the validation loss did not improve for 50 epochs. Rank convergence was then determined with a patience of 10 epochs and based on the Explained Variance Score, which was more robust for the audio modality. In the second phase, the model with its now-fixed ranks was fine-tuned for an additional up to 1000 epochs using a lower learning rate of $10^{-5}$. This fine-tuning stage utilized early stopping with a patience of 50 epochs based on the validation loss to ensure optimal performance.

#### A.8.6   MM-IMDB

We trained with the same hyperparameters as for the multi-modal simulations, but also here changed the distortion metric to the Explained Variance Score for stability in the text embeddings.

### A.9   ROBUSTNESS EXPERIMENTS

#### A.9.1   ROBUSTNESS TO DATA CHARACTERISTICS

We tested the robustness of our approach with respect to key properties of the simulated data: number of training samples, generative variable distribution, connectivity of generative variables, depth

and function of nonlinearity, and noise and dropout. To conduct these experiments, we utilized our parametric uni-modal simulation **A** to generate a suite of synthetic datasets. Starting from a default data configuration, we systematically varied one parameter at a time while holding all others constant to isolate its effect on our model's performance. We trained each configuration with the default simulation training setup described in A.8 for 5 random seeds. All tested values and results are shown in Figure 1.

### A.9.2 SAMPLE SIZE REQUIREMENTS

The number of samples (N) required for FiGuRO to produce a reliable ID estimate is not an absolute value but is instead dependent on the complexity of the autoencoder's encoder network. This relationship can be understood through a sample-to-capacity ratio, $\alpha$, which relates the number of samples to the number of encoder parameters $\mathcal{C}_e$ per latent dimension $k$ as $\alpha = \frac{kN}{\mathcal{C}_e}$ (Schuster & Krogh, 2021). In our experiments, we observed that robust estimates for the ground truth ID were achieved for sample sizes of N$\geq$5000. For the specific encoder architecture used in these tests, this corresponds to a sample-to-capacity ratio $\alpha \approx 20$. Since this autoencoder had a width ratio of 1 instead of 0.5 due to the small ambient dimension, we expect the $\alpha$ to be closer to 10. This finding is consistent with prior work on autoencoders, which suggests that a ratio of $\alpha \geq 10$ is generally required to sufficiently constrain the model (Schuster & Krogh, 2021).

### A.9.3 DISTORTION METRIC COMPARISON

We also tested the robustness of different general options for distortion metrics under varying nonlinearities and dropout levels. We tested $R^2$, MSE, RMSE, Explained Variance Score, and McFadden$R^2$ as potential distortion metrics. $R^2$, Explained Variance Score, and McFadden$R^2$ could be used with the absolute distortion threshold $\lambda = 0.05$ we introduced in the methods since they are supported in the range of $[0, 1]$. The remaining metrics we used with relative distortion thresholds, where we compute the maximum distortion as $1.05$ times the initial metric value. The nonlinearity types we tested were $x^2, \max(0, x), \frac{1}{1+e^{-x}}, \sin(x)$. We trained each configuration with the default simulation training setup described in A.8 for 5 random seeds. Figure 2 shows the average deviation from the true ID per metric and varied parameter.

## A.10 BASELINE METHOD IMPLEMENTATION

### A.10.1 CLASSICAL METHODS

We compare our approach against several non-neural network methods for ID estimation. These methods range from global linear techniques to local, neighbor-based estimators. We primarily utilized implementations from the `scikit-learn` and `skdim` python libraries, testing each method across a range of its key hyperparameters to ensure a fair and robust comparison.

- **Linear, Global**:
    - **Principal Component Analysis (PCA)**: ID is estimated as the number of components needed to explain a certain amount of variance. We used the `scikit-learn` implementation with variance `threshold` values of $\{0.8, 0.9, 0.95\}$ for the variance parameter $\sigma$.
    - **Singular Value Decomposition (SVD)**: The ID is estimated directly as the matrix rank, computed via `PyTorch` for explained energy ratios $EE \in \{100, 95, 90, 80\}\%$.
- **Random Matrix Theory**:
    - **BEMA (Bulk Edge Marchenko-Pastur Analysis)** (Ke et al., 2021): ID is the number of eigenvalues ("spikes") that exceed the theoretical upper bound of the Marchenko-Pastur distribution. We tested `bulk_percentiles` $\%ile \in \{80, 90, 95, 99\}$ to define the noise region.
- **Local, Neighbor-Based**:
    - **Local PCA (lPCA)** (Kambhatla & Leen, 1997): A variant of PCA that averages estimates from local neighborhoods. Tested with `alphaFO` values of $\{0.0001, 0.001, 0.01, 0.05, 0.1, 0.5, 0.9\}$ for parameter $\alpha$.

- **Correlation Integral (CorrInt)** (Camastra & Vinciarelli, 2002): A fractal dimension estimator. We performed a grid search over neighbor parameters $k'_1$ and $k'_2$ from the set $\{2, 5, 10, 20, 50, 100\}$.
- **FisherS** (Albergante et al., 2019): An estimator based on the separability of classes, run with its default parameters.
- **MiND_ML** (Rozza et al., 2012): A method based on the statistics of distances between nearest neighbors. Tested with the number of neighbors `k'` set to $\{2, 5, 10, 20, 50, 100\}$.
- **Maximum Likelihood Estimator (MLE)** (Levina & Bickel, 2004): A widely used estimator based on nearest-neighbor distances. We performed a grid search over the noise parameter `sigma` in $\{0, 0.001, 0.01, 0.1\}$ and the number of neighbors `K` in $\{2, 5, 10, 20, 50, 100\}$.
- **TLE** (Amsaleg et al., 2022): An estimator based on the two-sample log-likelihood of nearest neighbor distances. Tested with `epsilon` values of $\{10^{-10}, 10^{-5}, 10^{-4}, 10^{-3}, 10^{-2}, 0.1, 1.0\}$.
- **TwoNN** (Facco et al., 2017): An estimator based on the ratio of distances to the first and second nearest neighbors. Tested with `discard_fraction` $\frac{r_2}{r_1}$ values of $\{0.0, 0.1, 0.2, 0.3, 0.5, 0.7, 0.9\}$.

### A.10.2 NEURAL-NETWORK METHODS

We implemented a small number of NN-based techniques that can be used for ID estimation. Their training procedures are described below.

**Loss cliff** (Bahadur & Paffenroth, 2020): What we refer to as the loss cliff is a standard way of using autoencoders to roughly estimate the minimum bottleneck capacity an autoencoder (AE) needs to effectively reconstruct the data described by Bahadur & Paffenroth (2020). The core idea is that the reconstruction error will remain low as the latent dimension, $k$, is reduced, until $k$ drops below the true ID, at which point the error increases sharply, forming a "cliff" or "elbow". We employ the same autoencoder as four our method except from the decomposition layer. The training objective and training hyperparameters are the same as for our method, described in A.8. To find the cliff, we define an upper and lower limit for $k$ and train autoencoders with those bottlenecks. Instead of testing many dimensions, we search for 20 steps, evaluating the midpoint bottleneck between the lower and upper bound. Bound bottlenecks are dynamically updated based on whether the reconstruction loss was above or below the threshold $\mathcal{L}_{min} \times (1 + \lambda')$. We stop as soon as the range is $\leq 10$ epochs. This way we narrow down the range of the ID, but it requires training up to 21 models. We repeat this for three random seeds.

**Rank Reduction Autoencoder** (Mounayer et al., 2025): We implemented the Rank Reduction Autoencoder (RRA) from Mounayer et al. (2025) with a slight modification learning the weights as matrix decompositions as in our proposed approach. Ranks are pruned based on the rank reduction threshold $\gamma$ until no more singular values are above $\gamma$. We use the same architecture and training hyperparameters as for our approach and test thresholds for $\gamma \in [0.0001, 0.1]$. We again repeat the experiment for the same three random seeds as the other methods.

**ARD-VAE** (Saha et al., 2025): Another NN-based method we compare to is the ARD-VAE, a Variational Autoencoder (Kingma & Welling, 2022) with an Automatic Relevance Determination (ARD) prior (Saha et al., 2025). We implemented it according to the setup described below. However, VAEs collapsed for some random seeds even with low beta terms, potentially due to the high latent dimension and capacity of the networks we needed to train.

We again use the same architecture and training hyperparameters as for our approach, but add additional latent dimensions to model not just $\mu$ but also $\sigma^2$ for the approximate posterior $q(z|x) = \mathcal{N}(z|\mu, \text{diag}(\sigma^2))$, which latents $z$ are sampled from. Our implementation is based on the principles outlined by the original authors but uses a corrected loss function to ensure mathematical validity. The key to this method is the hierarchical ARD prior placed on the latent variables, where $\boldsymbol{\alpha}$ is a vector of learnable precision parameters. During training, the model optimizes the evidence lower bound (ELBO) (Kingma & Welling, 2022), which includes a Kullback-Leibler (KL) divergence term between the approximate posterior and the ARD prior:

$$\mathcal{L} = \mathbb{E}_{q(z|x)}[\log p(x|z)] - D_{KL}(q(z|x)||p(z|\boldsymbol{\alpha})) \tag{7}$$

This objective encourages the model to prune uninformative dimensions by driving their corresponding precisions $\alpha_i$ towards zero (i.e., their variance towards infinity). After training, the intrinsic dimension was determined by the relevance score. The relevance score $\hat{\sigma}_{\mathbf{w}}^2 = \mathbf{w}_{\hat{\sigma}} \odot \hat{\sigma}^2$ with weight vector $\mathbf{w}_{\hat{\sigma}}$ based on the Jacobian is better suited to find the relevant dimensions than defining a threshold of variance according to Saha et al. (2025). They determine the relevant dimensions based on 99% explained variance in $\hat{\sigma}_{\mathbf{w}}^2$.

### A.10.3 MULTI-MODAL DATA DECOMPOSITION METHODS

To evaluate the performance of our proposed model in in the multi-modal setting, we compare it against five baseline methods, although these baselines assume linear mixing of the hidden variables and some do not give estimates on the individual spaces (CCA, DIVAS). These methods range from classical correlation techniques to modern spectral decomposition algorithms, each designed to separate shared (joint) and modality-specific (individual) variation from two data views, $X_1 \in \mathbb{R}^{N \times n_1}$ and $X_2 \in \mathbb{R}^{N \times n_2}$. For all methods, data views are first centered by subtracting the feature-wise mean. Where applicable, initial signal ranks are estimated using the Optimal Hard Threshold (OHT) method, which is based on Random Matrix Theory (Gavish & Donoho, 2014) and applied in PPD.

**CCA (Canonical Correlation Analysis)** (Hotelling, 1936): CCA finds a shared subspace by identifying pairs of basis vectors (one for each view) that maximize the correlation between the projected data. In our implementation, the joint rank $k_J$ is determined by the number of components required to explain 80% of the cumulative squared canonical correlations (the correlation "energy"), which performed best in the uni-modal SVD baseline.

**DIVAS (Concatenated SVD)** (Prothero et al., 2024): We implement DIVAS as an "early fusion" baseline. The centered data matrices are concatenated, $Z = [X_1, X_2]$. A single Singular Value Decomposition (SVD) is performed on $Z$, and the joint rank $k_J$ is estimated using the OHT method (Gavish & Donoho, 2014) on $Z$'s spectrum. The joint components $(J_1, J_2)$ are reconstructed from this rank-$k_J$ approximation. This method assumes all non-joint signal is noise and does not model individual subspaces.

**JIVE (Joint and Individual Variation Explained)** (Lock et al., 2013): This method first estimates the individual signal ranks $(k_1, k_2)$ using OHT. It then performs SVD on the concatenated *signal bases* $(Z_{\text{basis}} = [U_1, U_2])$ and estimates the joint rank $k_J$ from $Z_{\text{basis}}$'s spectrum, again using OHT. The joint basis is used to project the signal matrices, yielding $J_1$ and $J_2$, with the residuals forming the individual components $I_1$ and $I_2$.

**AJIVE (Angle-based JIVE)** (Feng et al., 2018): We use the AJIVE implementation from the `py-jive` library (Feng et al., 2018). AJIVE also performs a full decomposition but uses a more robust procedure for rank estimation. It determines the joint rank $k_J$ by analyzing the principal angles between signal subspaces, using a permutation-based test to distinguish shared from non-shared variation. This process is stochastic, and we use a fixed random seed for reproducibility.

**PPD (Product of Projections Decomposition)** (Sergazinov et al., 2025): This spectral technique also begins by estimating signal ranks $(k_1, k_2)$ and projections $(P_1, P_2)$ via OHT. It then analyzes the spectrum of the *product of projections* matrix, $M = P_1 P_2$. The joint rank $k_J$ is estimated as the number of singular values of $M$ that exceed a dual threshold: (1) a perturbation bound $(1 - \epsilon_1)$ estimated via rotational bootstrap, and (2) a noise bound $(\lambda_+)$ derived from Random Matrix Theory. The joint subspace is estimated from a symmetric product matrix, and individual subspaces are defined as the remaining signal in the orthogonal complement of the joint space.

**SLIDE (Structural Learning and Integrative Decomposition)** (Gaynanova & Li, 2017): SLIDE is a decomposition method that relies on iterative optimization to enforce a sparse, block structure on the factor loading matrix. After estimating the total signal ranks $(k_1, k_2)$ for each modality via OHT and estimating the joint rank $(k_J)$ using the concatenation method (similar to JIVE), SLIDE iteratively updates the projection matrices. This process minimizes the reconstruction error while ensuring the loadings adhere to a predefined, structured sparsity pattern that separates the joint and individual variation. This yields explicit reconstructions for the joint components $(\mathbf{J}_1, \mathbf{J}_2)$ and individual components $(\mathbf{I}_1, \mathbf{I}_2)$.

**ShIndICA (Shared and Individual Independent Component Analysis)** (Pandeva & Forré, 2023): ShIndICA is a method that decomposes the data based on statistical independence (a principle derived from Independent Component Analysis, ICA), rather than orthogonal variance (like SVD-based methods). The goal is to find sources ($\mathbf{Z}$) that are maximally non-Gaussian. The decomposition is achieved by maximizing the non-Gaussianity of the source signals while enforcing a penalty that aligns the shared sources across modalities. Crucially, ShIndICA implements an automatic model selection procedure by testing a range of possible joint ranks and selecting the one that minimizes the Normalized Reconstruction Error (NRE) on a held-out test split. This means that the method needs to be run over various settings. We tested joint rank options $[1, 5, 10, 20]$.

### A.11 REAL DATASETS

#### A.11.1 NINFEA DATA

**Dataset access and description:** The "Non-Invasive Multimodal Foetal ECG-Doppler Dataset for Antenatal Cardiology Research (NInFEA)" dataset (Sulas et al., 2021) is available under Open Data Commons Attribution License on PhysioNet (doi: 10.13026/c4n5-3b04). It consists of 60 recordings from 39 pregnant women between the 21st and 27th week of gestation. Each record contains multiple synchronized signals. For our experiments, we utilized the following key modalities described in the dataset :

- Abdominal Electrophysiology (fECG view): 24 unipolar channels recorded from the maternal abdomen and back at 2048 Hz. These signals contain the target fetal ECG (fECG) signal heavily mixed with the maternal ECG (mECG) and other noise.

- Thoracic Electrophysiology (mECG view): 3 bipolar channels from the maternal thorax, primarily capturing the maternal ECG for reference.

- Maternal Respiration: A single channel from a piezo-resistive respiration belt, sampled at 2048 Hz.

- Pulsed-Wave Doppler (PWD): Provided as a single wide image in bitmap (.bmp) format, representing the Doppler velocity spectrum of the mechanical fetal heart activity.

**Data preprocessing:** The recordings were of different lengths, so we cropped ECG and respiration signals to the first 15351 data points per channel. Before cropping PWD images, we aligned them to be centered around the baseline (horizontal middle line of the ultrasound) and cropped them to $263 \times 2128$ pixel with three channels. We normalized all modalities to a range of $[0, 1]$ per channel.

**Ethical Considerations:** As stated in the original publication, the dataset was collected with approval from the Independent Ethics Committee of the Cagliari University Hospital (AOU Cagliari) and all volunteers provided signed informed consent.

#### A.11.2 AUDIO MNIST

We used the Audio MNIST dataset from (Becker et al., 2023), which provided 5000 train and 1000 test samples per digit derived from pairing MNIST (Deng, 2012) and FSDD (Jackson et al., 2018). MNIST images are black and white images of size $24 \times 24$ and audio samples are spectrograms of size $112 \times 112$.

#### A.11.3 MM-IMDB

We used the MM-IMDB dataset from (Arevalo et al., 2017), which provides 15,552 train, 2,608 validation, and 7,799 test samples derived from pairing MovieLens 20M metadata with movie posters and plot summaries. Text inputs are processed using pre-trained Word2Vec (Mikolov et al., 2013) embeddings of size 300, and 4096 image features were extracted using VGG-16 (Simonyan & Zisserman, 2014). This processing was done by Liang et al. (2021), from where we downloaded the data.

### A.12 LLM USAGE

LLMs were used to polish text and supplement our manual literature research. They were also used to assist with code. Ideation, results, and the content of the paper were solely contributed by the authors.

# B  TABLES

Supplementary Table 1: **FiGuRO Hyperparameter sensitivity analysis.** We report the mean estimated rank and mean deviation from the ground truth (GT) ID ($\pm$ SEM) across all simulated datasets and random seeds for the given number of samples (N). Bold parameter values indicate chosen hyperparameters, bold ranks show best results based on rank and deviation (if applicable).

| Hyperparameter | Value | Estimated Rank | Deviation from GT | Sample size (N) |
|---|---|---|---|---|
| Distortion threshold ($\lambda$) | 0.005 | $6.56 \pm 0.04$ | $1.65 \pm 0.04$ | 180 |
| | 0.01 | $6.16 \pm 0.04$ | $1.29 \pm 0.04$ | |
| | **0.05** | $\mathbf{5.02 \pm 0.01}$ | $\mathbf{0.50 \pm 0.01}$ | |
| | 0.1 | $4.45 \pm 0.01$ | $0.68 \pm 0.01$ | |
| | 0.15 | $3.94 \pm 0.00$ | $1.06 \pm 0.00$ | |
| | 0.2 | $3.57 \pm 0.01$ | $1.43 \pm 0.01$ | |
| Frequency ($\tau$) | 5 | $5.24 \pm 0.02$ | $1.17 \pm 0.02$ | 360 |
| | **10** | $\mathbf{4.81 \pm 0.02}$ | $\mathbf{1.02 \pm 0.02}$ | |
| | 20 | $4.48 \pm 0.02$ | $1.11 \pm 0.02$ | |
| Energy threshold $\gamma$ | 0.0001 | $\mathbf{4.99 \pm 0.02}$ | $1.29 \pm 0.02$ | 270 |
| | 0.001 | $5.11 \pm 0.03$ | $1.16 \pm 0.02$ | |
| | **0.01** | $4.87 \pm 0.03$ | $1.09 \pm 0.03$ | |
| | 0.1 | $4.57 \pm 0.02$ | $\mathbf{0.85 \pm 0.01}$ | |
| Patience ($\pi$) | 5 | $5.30 \pm 0.04$ | $1.23 \pm 0.04$ | 216 |
| | **10** | $4.73 \pm 0.02$ | $\mathbf{0.88 \pm 0.02}$ | |
| | 20 | $5.06 \pm 0.03$ | $1.08 \pm 0.03$ | |
| | 50 | $\mathbf{5.03 \pm 0.03}$ | $1.30 \pm 0.03$ | |
| | 100 | $4.62 \pm 0.01$ | $1.07 \pm 0.01$ | |

---

[1]Two out of three random seeds resulted in posterior collapse and ID estimates of 1 for $\mathbf{D_2}$. For $\mathbf{D_1}$, KL weights $\beta \leq 0.0001$ resulted in posterior collapse and an ID estimate of 1.

Supplementary Table 2: **Unimodal ID estimation benchmark.** We compare FiGuRO to a range of statistical estimators (Global, Local) and other neural network-based (NN) methods. nn refers to nearest neighbor. Ranges indicate estimates for varying hyperparameters (if applicable, in brackets next to the method). The respective hyperparameters and ranges tested for each method are described in A.10. Neural network-based methods are reported as means from three random seeds with SEM.We round the best estimate for each method and indicate best overall estimates compared to the ground truth ID with bold numbers, second best underlined.

| Category | Method | $D_1$ (ID = 11) | $D_2$ (ID = 12) | Best ($D_1$,$D_2$) |
|---|---|---|---|---|
| Global | SVD ($CE$) | 1522 - 9424 | 1 - 104 | 1522, 1 ($CE = 80\%$) |
| | PCA ($\sigma^2$) | 4040 - 9283 | 12 - 29 | 4040, 12 ($\sigma^2 = 0.8$) |
| | BEMA (%$ile$) | 570 - 948 | 273 - 1322 | 570, 273 (%$ile = 99$) |
| Local, proj. | lPCA ($\alpha$) | 4 - 8862 | 2 - 86 | **10, 9** ($\alpha = 0.05$) |
| | FisherS | 3.4 | 1.8 | 3, 2 |
| Local, geom. | CorrInt ($k_1'$, $k_2'$) | 5.2 - 5.6 | 2.9 - 3.3 | 6, 3 ($k_1' = 2, k_2' = 5$) |
| | TLE | 44 | 2.9 | 44, 3 |
| Local, nn | MindML ($k'$) | 68 - 172 | 2.6 - 3.0 | 68, 3 ($k' = 100$) |
| | MLE | 53 | 2.4 | 53, 2 |
| | TwoNN ($\frac{r_2}{r_1}$) | 130 - 229 | 3.1 - 3.3 | 130, 3 ($\frac{r_2}{r_1} = 0$) |
| NN | Loss cliff ($\lambda'$) | $512.0 \pm 0.0$ - $728.0 \pm 0.0$ | $68.0 \pm 5.0$ - $258.7 \pm 188.2$ | $512.0 \pm 0.0, 68 \pm 5.0$ ($\lambda' = 0.8$) |
| | ARD-VAE ($\beta$)[1] | $1.0 \pm 0.0$ - $14.0 \pm 0.0$ | $1.0 \pm 0.0$ - $93.3 \pm 92.3$ | $14.0 \pm 0.6$ , $7.3 \pm 6.3$ ($\beta = 0.001$) |
| | RRA ($\gamma$) | $5.3 \pm 0.2$ - $146 \pm 0.6$ | $1.3 \pm 0.3$ - $138 \pm 6.0$ | $14.0 \pm 1.0, 8.7 \pm 0.7$ ($\gamma = 0.05$) |
| Ours | FiGuRO ($\lambda$) | **$10.4 \pm 0.2$ - $20.0 \pm 0.6$** | **$4.3 \pm 0.0$ - $11.7 \pm 0.9$** | $15.3 \pm 2.2, 5.7 \pm 0.3$ ($\lambda = 0.05$) |

Supplementary Table 3: **Ablation study.** This table compares the average estimated ranks ($k_s$, $k_1$, $k_2$) of a naive SVD-only rank reduction model against the full FiGuRO implementation (SVD + R(D) algorithm) and ground truth (GT) across datasets **B**. We report mean $\pm$ SEM on three random seeds.

| Method | Subspace | $B_s$ | $B_{i1}$ | $B_{i2}$ | $B_l$ |
|---|---|---|---|---|---|
| GT | $k_s$ | 2 | 20 | 2 | 20 |
| | $k_1$ | 3 | 2 | 2 | 20 |
| | $k_2$ | 5 | 2 | 20 | 20 |
| FiGuRO | $k_s$ | $1.0 \pm 0.0$ | $1.0 \pm 0.0$ | $1.0 \pm 0.0$ | $1.0 \pm 0.0$ |
| (SVD) | $k_1$ | $1.0 \pm 0.0$ | $1.0 \pm 0.0$ | $1.0 \pm 0.0$ | $1.0 \pm 0.0$ |
| | $k_2$ | $1.0 \pm 0.0$ | $1.0 \pm 0.0$ | $1.0 \pm 0.0$ | $1.0 \pm 0.0$ |
| FiGuRO | $k_s$ | $3.6 \pm 0.06$ | $13.4 \pm 0.4$ | $7.0 \pm 0.0$ | $19.2 \pm 0.7$ |
| (SVD + | $k_1$ | $1.0 \pm 0.0$ | $4.4 \pm 0.2$ | $1.0 \pm 0.0$ | $12.8 \pm 1.0$ |
| R(D)) | $k_2$ | $6.2 \pm 1.8$ | $4.0 \pm 0.5$ | $19.4 \pm 1.5$ | $15.2 \pm 1.3$ |

Supplementary Table 4: **ID estimation with $L_1$ regularization.** Comparison of the average estimated ranks ($k_s$, $k_1$, $k_2$) when adding varying weights of $L_1$ regularization to the latent spaces. We report only the mean for three random seeds.

| L1 weight | $B_s$ | $B_{i1}$ | $B_{i2}$ | $B_l$ |
|---|---|---|---|---|
| 0.00 | 3.6, 1.0, 6.2 | 13.4, 4.4, 4.0 | 7.0, 1.0, 19.4 | 19.2, 12.8, 15.2 |
| 0.01 | 1.0, 1.0, 2.3 | 9.7, 6.3, 3.3 | 1.0, 1.0, 6.0 | 14.0, 9.0, 14.3 |
| 0.10 | 1.0, 1.0, 2.0 | 10.3, 5.7, 4.3 | 1.0, 1.0, 17.7 | 14.3, 8.7, 15.3 |
| 1.00 | 1.0, 1.0, 2.0 | 10.0, 6.0, 3.0 | 1.0, 1.0, 9.7 | 14.0, 9.0, 15.0 |
| 10.00 | 1.0, 1.0, 2.7 | 10.3, 6.3, 2.0 | 1.0, 1.0, 10.0 | 14.0, 10.7, 14.3 |

Supplementary Table 5: **Disentanglement with $L_1$ regularization.** We report mean classification accuracy of label 0 accuracy from the shared subspace ($Acc_s$) and predictability ($R^2$) of label 1 and 2 from their respective subspaces in format ($Acc_s$, $R_1^2$, $R_2^2$) per $L_1$ weight and dataset (N=3).

| L1 weight | $\mathbf{B_s}$ | $\mathbf{B_{i1}}$ | $\mathbf{B_{i2}}$ | $\mathbf{B_l}$ |
|---|---|---|---|---|
| 0.00 | 1.00, 0.29, 0.94 | 0.98, 0.99, 1.00 | 1.00, 0.61, 0.98 | 0.99, 0.99, 0.99 |
| 0.01 | 0.74, 0.04, 0.92 | 0.94, 0.92, 0.97 | 0.75, 0.50, 0.77 | 0.91, 0.45, 0.82 |
| 0.10 | 0.75, 0.33, 0.96 | 0.94, 0.91, 0.98 | 0.83, 0.20, 0.83 | 0.90, 0.31, 0.86 |
| 1.00 | 0.82, 0.34, 0.94 | 0.94, 0.93, 0.97 | 0.79, 0.50, 0.96 | 0.90, 0.35, 0.84 |
| 10.00 | 0.88, 0.41, 0.94 | 0.94, 0.92, 0.97 | 0.73, 0.50, 0.64 | 0.91, 0.65, 0.85 |

Supplementary Table 6: **ID estimation with orthogonal loss (Frobenius norm).** Comparison of the average estimated ranks ($\mathbf{k_s}, \mathbf{k_1}, \mathbf{k_2}$) when adding varying weights to the latent spaces. We report only the mean for three random seeds.

| Orthogonal weight | $\mathbf{B_s}$ | $\mathbf{B_{i1}}$ | $\mathbf{B_{i2}}$ | $\mathbf{B_l}$ |
|---|---|---|---|---|
| 0.00 | 3.6, 1.0, 6.2 | 13.4, 4.4, 4.0 | 7.0, 1.0, 19.4 | 19.2, 12.8, 15.2 |
| 0.01 | 1.0, 1.0, 2.3 | 10.3, 6.3, 2.0 | 1.0, 1.0, 16.3 | 15.0, 8.3, 16.0 |
| 0.10 | 1.0, 1.0, 2.0 | 10.3, 6.3, 2.3 | 1.0, 1.0, 12.7 | 13.7, 7.7, 15.7 |
| 1.00 | 1.3, 1.7, 3.3 | 14.0, 2.0, 3.3 | 1.0, 1.7, 11.0 | 13.7, 9.0, 18.0 |
| 10.00 | 1.3, 1.7, 5.7 | 14.0, 5.3, 5.3 | 1.0, 1.7, 14.3 | 15.0, 12.7, 30.7 |

Supplementary Table 7: **Disentanglement with orthogonal loss (Frobenius norm).** We report mean classification accuracy of label 0 accuracy from the shared subspace ($Acc_s$) and predictability ($R^2$) of label 1 and 2 from their respective subspaces in format ($Acc_s$, $R_1^2$, $R_2^2$) per weight and dataset (N=3).

| Orthogonal weight | $\mathbf{B_s}$ | $\mathbf{B_{i1}}$ | $\mathbf{B_{i2}}$ | $\mathbf{B_l}$ |
|---|---|---|---|---|
| 0.00 | 1.00, 0.29, 0.94 | 0.98, 0.99, 1.00 | 1.00, 0.61, 0.98 | 0.99, 0.99, 0.99 |
| 0.01 | 0.74, 0.01, 0.35 | 0.94, 0.91, 0.96 | 1.00, 0.08, 0.83 | 0.90, 0.18, 0.86 |
| 0.10 | 0.84, 0.08, 0.49 | 0.94, 0.95, 0.95 | 0.77, 0.05, 0.84 | 0.89, 0.22, 0.86 |
| 1.00 | 0.50, 0.15, 0.58 | 0.94, 0.63, 0.97 | 0.25, 0.03, 0.98 | 0.90, 0.47, 0.78 |
| 10.00 | 0.47, 0.07, 0.89 | 0.94, 0.83, 0.98 | 0.44, 0.34, 0.94 | 0.89, 0.14, 0.81 |

Supplementary Table 8: **Edge case: Multi-modal data without shared information. $L_2$ regularization enforces disentanglement.** This table shows the effect of increasing the $L_2$ weight on the decoder's first layer when trained on a simulated dataset with **no true shared ID** ($k_{s,GT} = 0$) and modality-specific IDs of 20 (a modification of $\mathbf{B_l}$). A lower shared rank $k_s$ is better. $k_{tot}$ refers to the total rank (sum of all subspace ranks). Predictability of each modality's label is given by the goodness of fit $R^2$. This was done for three random seeds, showing the mean and SEM.

| L2 weight | $k_s \downarrow$ | $k_{tot}$ | $k_s/k_{tot} \downarrow$ | $R^2$ label 1 $\uparrow$ | $R^2$ label 2 $\uparrow$ |
|---|---|---|---|---|---|
| 0.00 | $9.7 \pm 0.3$ | $23.7 \pm 0.3$ | 0.40 | $0.40 \pm 0.14$ | $0.99 \pm 0.00$ |
| 0.01 | $4.3 \pm 1.9$ | $23.7 \pm 0.7$ | 0.18 | $0.69 \pm 0.20$ | $0.98 \pm 0.01$ |
| 0.10 | $7.0 \pm 5.0$ | $31.5 \pm 8.5$ | 0.22 | $0.74 \pm 0.25$ | $0.99 \pm 0.00$ |
| 1.00 | $2.7 \pm 0.7$ | $26.7 \pm 9.2$ | 0.10 | $0.99 \pm 0.01$ | $0.98 \pm 0.01$ |
| 10.00 | $1.0 \pm 0.0$ | $12.0 \pm 1.7$ | 0.08 | $0.97 \pm 0.01$ | $0.98 \pm 0.01$ |

Supplementary Table 9: **Information overlap between pairs of NInFEA modalities measured as the ratio of shared over total rank.** The total rank is calculated as the sum of all subspace ranks. "*" denotes a modality pair we initially expected to have a small overlap but showed strong technical bias from one modality onto the other.

|  | fECG-mECG | fECG-fPWD | mECG-mR | fECG-mR | mECG-fPWD | mR-fPWD |
|---|---|---|---|---|---|---|
| $k_s$ / $k_{tot}$ | **0.38** | **0.54** | **0.47** | 0.37* | 0.17 | 0.13 |

Supplementary Table 10: **Subspace IDs for FiGuRO with three modalities.** Trained on NInFEA.

| Subspace | ID |
|---|---|
| global shared | 3 |
| fECG + respiration | 3 |
| fECG + PWD | 8 |
| respiration + PWD | 5 |
| fECG | 1 |
| respiration | 1 |
| PWD | 1 |

Supplementary Table 11: **ID estimation on real multi-modal datasets.** We report ID estimates for shared ($k_s$) and modality-specific subspaces on different datasets and methods. Our NN-based method was evaluated on 3 random seeds, reporting the mean $\pm$ SEM. We excluded CCA and DIVAS as they do not estimate individual subspaces, and PPD because it failed to return predictions on Audio MNIST after 2 days of running. For our method, we used a distortion budget of $\lambda = 0.05$.

| Dataset | Subspace | JIVE | AJIVE | SLIDE | ShIndICA | **Ours** |
|---|---|---|---|---|---|---|
| Audio MNIST | $k_s$ | 1 | 9 | 1 | 1 | $9.7 \pm 0.3$ |
|  | $k_1$ | 133 | 135 | 117 | 117 | $1.7 \pm 0.3$ |
|  | $k_2$ | 230 | 485 | 234 | 234 | $5.0 \pm 0.6$ |
| MM-IMDB | $k_s$ | 1 | 13 | 1 | 1 | $19.0 \pm 1.2$ |
|  | $k_1$ | 708 | 695 | 707 | 707 | $1.0 \pm 0.0$ |
|  | $k_2$ | 65 | 55 | 64 | 64 | $61.3 \pm 3.4$ |

Supplementary Table 12: **Reconstruction performance before and after rank optimization.** $\mathcal{L}$ indicates the loss (MSE) for train and test set, with $\Delta$ indicating the difference in loss from initial ($t = 0$) to final rank ($\mathcal{L}^t - \mathcal{L}^0$). We report mean and SEM over three random seeds. $\%$ indicates the relative increase in loss. For final ranks we only show mean, for losses we show mean +- SEM. $k_0$ refers to the initial sum of max ranks we set for each subspaces, $k_{total}$ to the mean sum of all final ranks over the random seeds.

| Dataset | $k_0$ | $k_{total}$ | $\mathcal{L}_{train}$ | $\mathcal{L}_{test}$ | $\Delta\mathcal{L}_{train}$ | $\Delta\mathcal{L}_{test}$ | $\%_{train}$ | $\%_{test}$ |
|---|---|---|---|---|---|---|---|---|
| Audio MNIST | 600 | 16.4 | $0.6001 \pm 0.0025$ | $0.3153 \pm 0.0013$ | $0.0061 \pm 0.0010$ | $0.0007 \pm 0.0003$ | 10% | 0.2% |

## C FIGURES

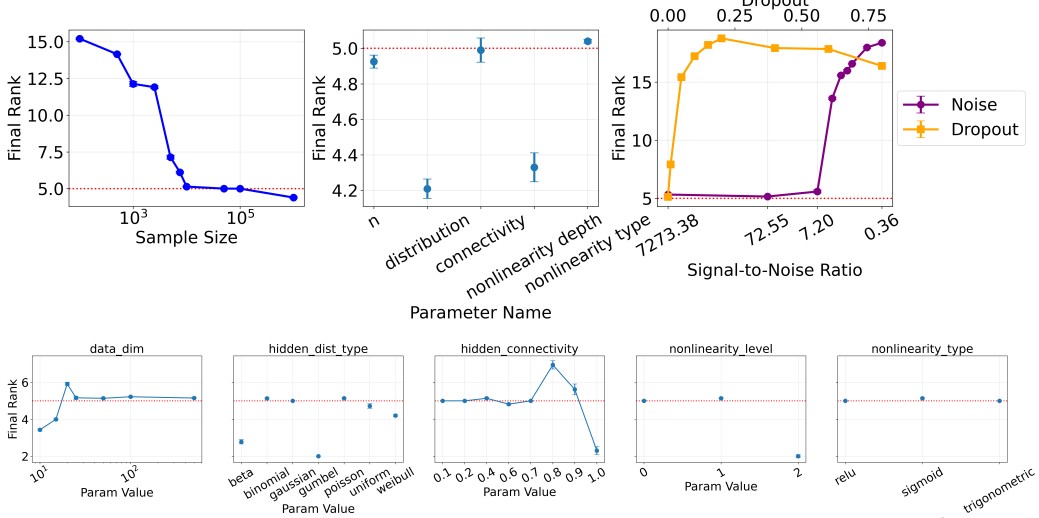

Supplementary Figure 1: **Robustness tests.** We evaluated the method's stability on simulated data (ground truth ID of 5, red dotted lines) by systematically varying key generative parameters and running for three random seeds. **Top row:** The top left plot shows that the estimated rank converges to the true ID as sample size increases, stabilizing around N=5000. The middle plot displays the mean final rank across the different data characteristics we tested. The top right plot demonstrates robustness to noise up to a high signal-to-noise ratio (SNR) but shows overestimation with high levels of dropout. **Bottom Row** These plots provide a more detailed view of the top middle plot. Each subplot represents one of the data characteristics from the x axis of the top left plot. Error bars indicate the SEM from three random seeds.

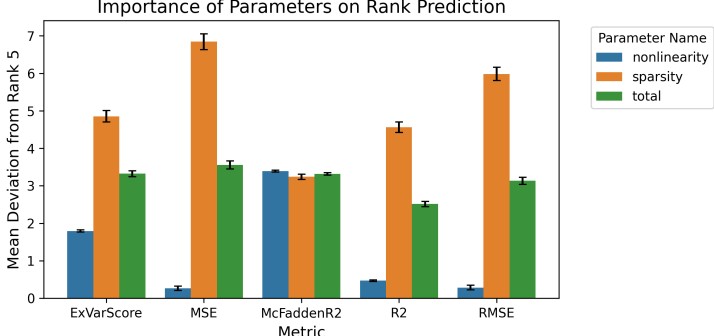

Supplementary Figure 2: **Distortion metrics.** We evaluated five different distortion metrics by measuring the mean deviation of their ID estimates from the ground truth (ID=5) across simulated datasets with varying levels of nonlinearity and sparsity. We report the average deviation from the ground truth ID 5 with error bars indicating the SEM from three random seeds.

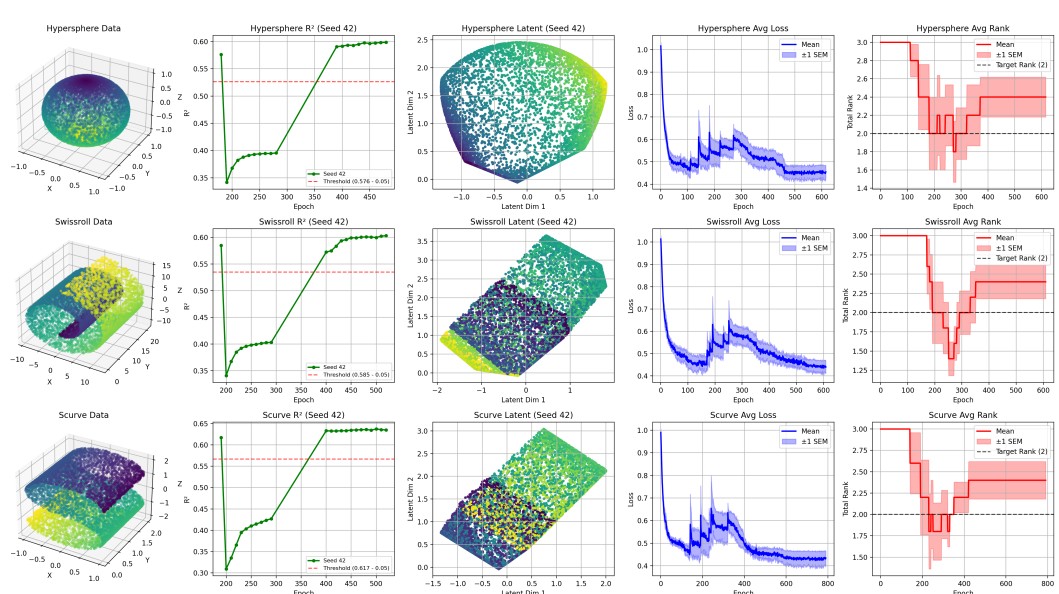

Supplementary Figure 3: **Training dynamics on 3D Manifold datasets.** Each row corresponds to a different dataset: Hypersphere, Swiss Roll, and S-Curve. For each dataset, the columns show: the original 3D data, the R² metric during rank optimization (red dashed line showing the distortion budget), the learned 2D latent representation, the average training loss over 5 seeds, and the average rank convergence over 5 seeds. The target rank of 2 is indicated by the dashed grey line.

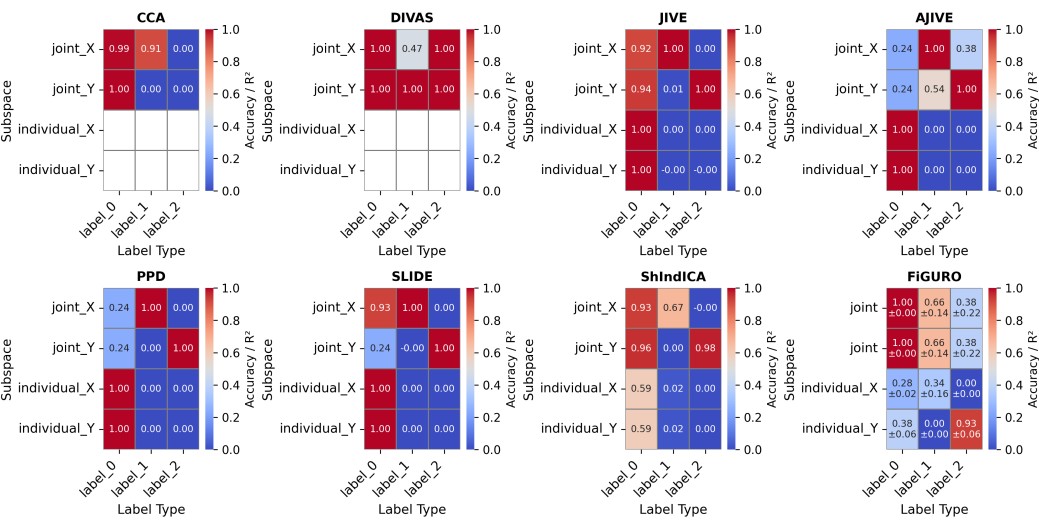

Supplementary Figure 4: **Disentanglement evaluation of multi-modal baselines on dataset $B_s$.** Each heatmap plots the predictability of the three ground truth labels (0: shared, 1: modality 1, 2: modality 2) from the decomposed joint and individual (private) subspaces per method. Predictability is evaluated as in Figure 2B as classification accuracy and $R^2$. For our method FiGuRO, the joint predictabilities are duplicated as there is only one learned joint subspace, and values are reported as mean $\pm$ SEM from 5 random seeds. Empty cells for the first two methods in individual components indicate that these methods did not estimate individual subspaces.

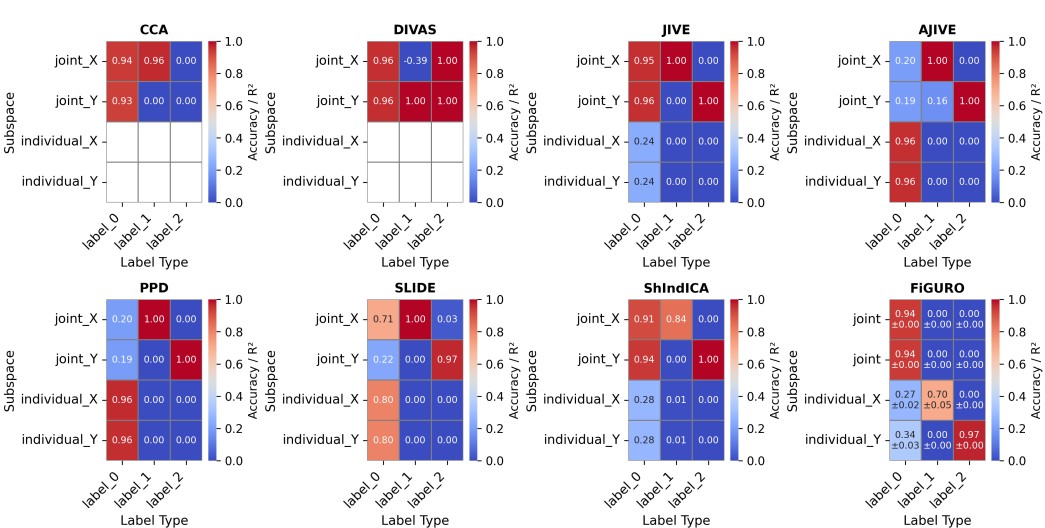

Supplementary Figure 5: **Disentanglement evaluation of multi-modal baselines on dataset $B_{i1}$.** For details see caption 4.

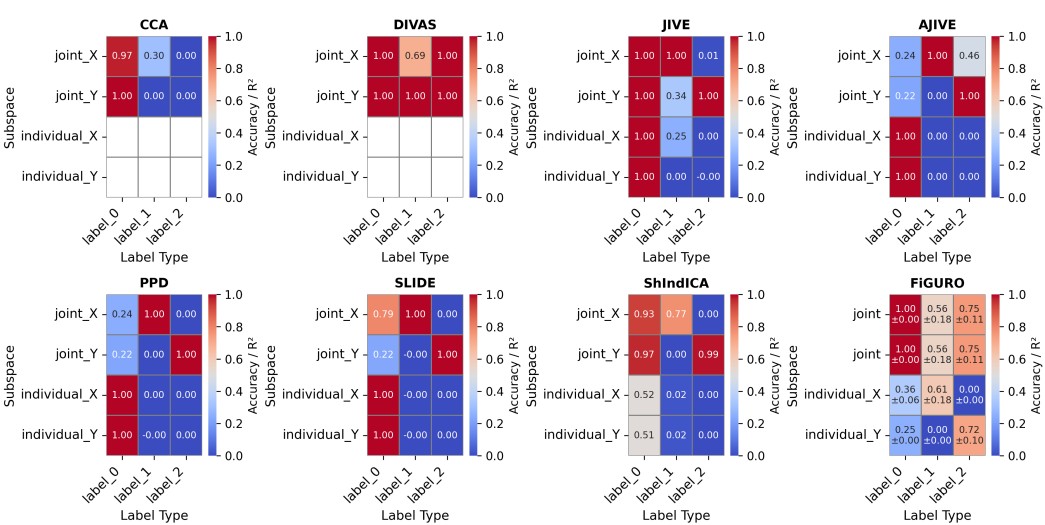

Supplementary Figure 6: **Disentanglement evaluation of multi-modal baselines on dataset $B_{i2}$.** For details see caption 4.

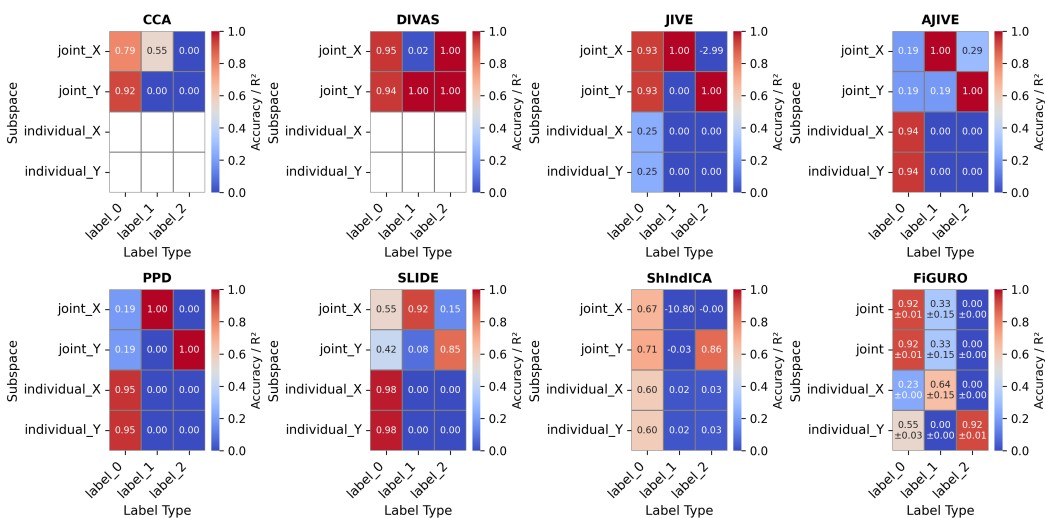

Supplementary Figure 7: **Disentanglement evaluation of multi-modal baselines on dataset** $B_1$. For details see caption 4.

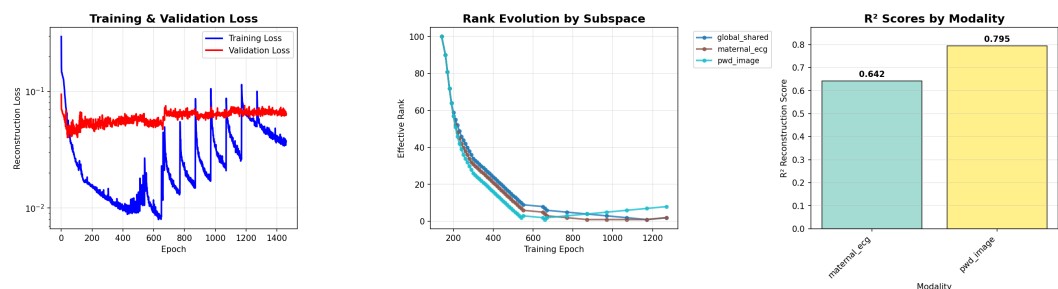

Supplementary Figure 8: **Training dynamics on NInFEA mECG-fPWD.** The left plot shows the training and validation loss. The middle plot depicts the ranks of all three subspaces over epochs. The right plot shows the initial $R^2$ metrics per modality.

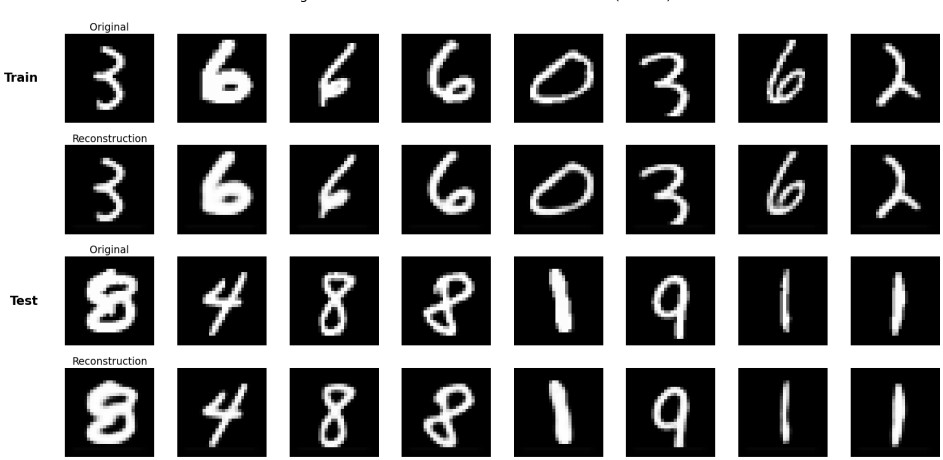

Supplementary Figure 9: **Image reconstruction samples from Audio MNIST.** The top row presents original test samples, the bottom its reconstructions from our pretrained Audio MNIST model (seed 0).

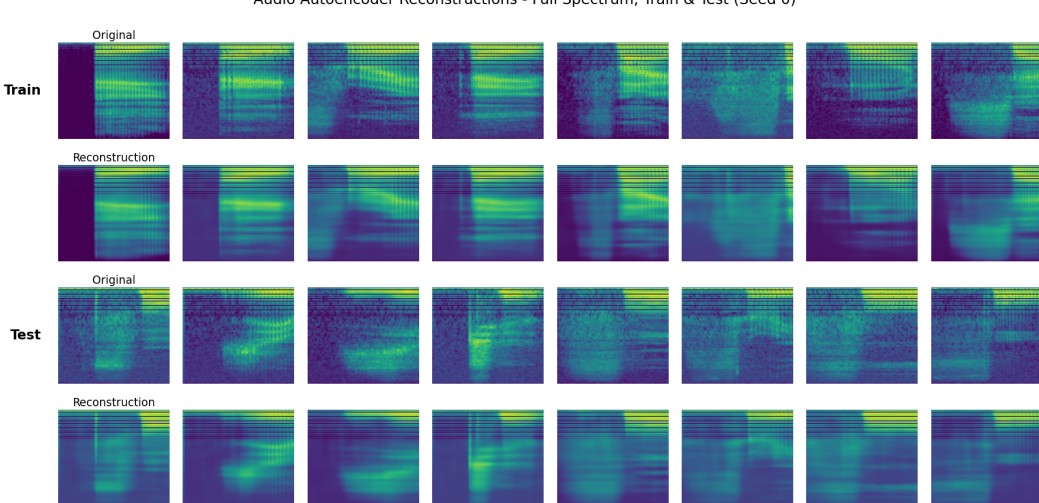

Supplementary Figure 10: **Audio reconstruction samples from Audio MNIST.** The top row presents original test samples, the bottom its reconstructions from our pretrained Audio MNIST model (seed 0).

