# OpenReview forum: "FiGuRO - Intrinsic Dimension Estimation for Multi-Modal Data"
_ICLR.cc/2026/Conference — Submitted to ICLR 2026_

### Official Review · Reviewer_y7sV · 2025-10-20

**Soundness:** 3
**Presentation:** 3
**Contribution:** 3
**Rating:** 4
**Confidence:** 4

**Summary:**

The authors propose a method to estimate the intrinsic multimodal data dimension: shared and source-specific. To achieve this, they rely on paired autoencoders and iteratively adapt their latent dimension size by learning a low-dimensional projection. Experiments on the simulated and real data show utility of the method.

**Strengths:**

1. **Novel methodology.** The proposed method of greedily adapting the latent dimension rank (rate) to achieve the desired error (distortion) is novel and appears justified.
2. **Clear motivation and writing.** The paper is mostly clear and well-explained. The problem is well-motivated.

**Weaknesses:**

1. **Lack of comparisons with multi-view data decomposition methods.** On line 288, the authors claim lack of "multi-modal ID estimation techniques," which is not strictly speaking true. There a vast literature on the multi-view data decomposition with canonical methods such as CCA [1], JIVE [2], AJIVE [3], DIVAS [4], and PPD [5]. While these methods assume a specific linear model of joint and individual mixing, I believe they provide a strong baseline that any non-linear methods must compare with. Moreover, unlike the current method which only estimates the joint and individual ranks, the methods [1-5] also produce estimates of the joint and individual subspaces (the spanning sets).  I believe this a critical omission in the current work.


2. As the original **motivating examples**, which are plausible, the authors list the following points why ID estimation is useful:
> (i) performance of deep neural networks has been shown to depend on the intrinsic rather than ambient dimension
> (ii) need interpretable models
> (iii) want to know whether expensive or difficult-to-obtain modalities are relevant

None of them are sufficiently illustrated in the experiments section. The authors apply their method, obtain the ID of each data source, and then check if it agrees with their intuition.

3. **Corner (degenerate) cases.** I feel the method crucially assumes that $k_s > 0$. In practice, the practitioners may be interested in applying the method exactly under the setup, where the dependence (shared) signal between the modalities is uncertain. However, from the examination of the algorithm steps, I believe the method would fail to converge in such case.

The exact setup I have in mind is supposing, we have Assumptions 1-4 hold: oracle autoencoders and correct data generating mechanism. But further suppose that $k_s = 0, k_1 = 10, k_2 = 10$, i.e., there is no joint. Going through the steps in Algorithm 1, because the updates to the individual / joint ranks are combined together from Eqn. 4 and Lines 13-14 in the algorithm, the problem becomes unidentifiable. I can see two distinct fixed point between which the algorithm is going to oscillate, estimate the joint to be rank $=20$ or two individual of rank $=10$ each. Namely, I don't see any mechanism by which the algorithm is supposed to correctly identify which option to prefer.

---
[1] Hotelling, H. (1936). Relations between two sets of variates. Biometrika, 28, 321–377.

[2] Lock, Eric F., et al. "Joint and individual variation explained (JIVE) for integrated analysis of multiple data types." The annals of applied statistics 7.1 (2013): 523.

[3] Feng, Qing, et al. "Angle-based joint and individual variation explained." Journal of multivariate analysis 166 (2018): 241-265.

[4] Prothero, Jack, et al. "Data integration via analysis of subspaces (DIVAS)." TEST 33.3 (2024): 633-674.

[5] Sergazinov, Renat, Armeen Taeb, and Irina Gaynanova. "A spectral method for multi-view subspace learning using the product of projections." arXiv preprint arXiv:2410.19125 (2024).

**Questions:**

1. What's the purpose of $k_{min}/k_{max}$, I don't see them being used anywhere in the algorithm.
2. On line 17 in the algorithm, it says "set ranks to their previous values". What are the initial values for these? Let's say the algorithm just started, then supposedly, the ranks have to be initialised already?
3. There are few typos: (i) on 188 "perform do" (ii) line 469 "Alltogether". The paper needs to be proof read.

---

> ### Author Response · Authors · 2025-11-21
> **(1/2)**
>
> We thank the reviewer for their thoughtful and expert review. We are encouraged that the reviewer found our methodology novel and our writing clear. We are especially grateful for the references regarding multi-view data decomposition. Incorporating this perspective has enriched our discussion of related work, significantly improved our results, and clarified our method’s utility. Below, we address the theoretical concerns regarding degenerate cases and discuss our additional findings.
>
> > **W1: Lack of comparisons with multi-view data decomposition methods**
>
> Thank you for pointing us towards this literature of linear multi-view data decomposition. You are absolutely correct that some of them could serve as good baselines and should be included. While CCA and DIVAS seem to only find the joint rank and subspace, we found two additional methods, SLIDE [1] and ShINDICA [2], that we added to our analysis along with JIVE and AJIVE. While PPD is theoretically interesting, its computational cost made it infeasible (timed out after 48h on Audio MNIST).
>
> Evaluating the performance of these baselines, we observed failure modes exposing clear weaknesses. The linear baselines swapped joint and individual information. We believe this could be due to such methods assuming the joint signal is the "strongest" or "most aligned" signal. Our method, on the other hand, correctly learns the joint information in the shared subspace, though there is some leakage from individual information into the shared space. Overall, our method achieved the best disentanglement and rank estimates (new Table 2, Supplementary Figures 4-7). The new Supplementary Figures include heatmaps of predictability of shared and modality-specific information from baseline decompositions, clearly showing these failure modes while FiGuRO mostly correctly splits the information.
>
> We would also like to address a misunderstanding in comment 1, and apologize if it was not made clear in the paper. Our method not only estimates the IDs/ranks of the shared and modality-specific subspaces. It DOES learn these latent subspaces themselves. This information decomposition has been demonstrated in Figure 2B, and we show learned subspaces in Figures 1 and 3B. It is now also further emphasized in the additional downstream task results in Section 4.2.2 and we added a note in the abstract (line 023f) to make it clear from the start.
>
> [1] Irina Gaynanova and Gen Li. Structural Learning and Integrative Decomposition of Multi-View
> Data, July 2017.
>
> [2] Teodora Pandeva and Patrick Forr´e. Multi-View Independent Component Analysis with Shared and
> Individual Sources. In Proceedings of the Thirty-Ninth Conference on Uncertainty in Artificial
> Intelligence, pp. 1639–1650. PMLR, July 2023.
>
> > **W2: Motivating examples**
>
> We thank the reviewer for appreciating the plausible motivating examples. We respectfully maintain that while we do not perform exhaustive studies on all potential downstream impacts, our experiments provide evidence illustrating the utility of the ID estimates for points (ii) and (iii). We provide further evidence with our new results on multi-modal real datasets.
>
> **(i)** We rely on literature to support point (i)—performance dependence on ID, as exhaustively proving this relationship for every dataset is outside the scope of introducing a novel methodology.
>
> **(ii)** Need for interpretable models: We see disentanglement of the inferred subspaces as a direct mechanism on the path to interpretability. Our results in Figure 2B and Figure 3B visually and quantitatively demonstrate this by showing how the shared subspace captures high classification accuracy (semantic/digit identity) while the private subspaces capture modality-specific noise or style (speaker voice/image variations). We have expanded our multi-modal real data experiments to demonstrate the downstream utility of this disentanglement (Section 4.2.2 lines 483ff, Supplementary Tables 11,12).
>
> **(iii)** Relevance of Modalities: The relevance of a modality is directly tied to the size and content of its estimated ID. We illustrate this in two ways:
> - Quantification of Contribution: Our analysis on the real-world NInFEA dataset (former Table 3, now Supplementary Table 9) validates clinical intuition by quantifying the information overlap ($k_s/k_{tot}$) between modalities. Modality pairs with known high correlation (e.g., fECG and fPWD) show the largest shared ID fraction and thus a high overlap and potential redundancy.
> - Probing the Space: The estimated modality-specific ID ($k_m$) serves as a foundational metric of relevance. A lower estimated $k_m$ (especially relative to other subspaces) strongly suggests that the modality contributes less unique information. Pairing this with downstream task prediction, one can quantify the task-specific relevance of each subspace, as we demonstrated in our multi-modal real data applications (Section 4.2.2).

---

> ### Author Response · Authors · 2025-11-21
> **(2/2)**
>
> > **W3: Corner cases**
>
> We appreciate the reviewer raising this excellent question regarding the corner case where there is no shared signal ($k_s = 0$). We agree that in the strict case of $k_s=0$, the solution space is unidentifiable without constraints.
>
> - Practical vs. Theoretical Assumption: We initially assumed $k_s > 0$, as multi-modal data is typically collected under the premise of some shared relationship (or metadata tie). We have added this assumption to Appendix A (new Assumption 2).
> - Testing the Edge Case: We tested the specific edge case of no shared information. As predicted, the model does not robustly converge to the true state ($k_s \approx 0$), often showing unstable results where one modality collapses and the shared rank is overestimated.
> - Introducing a regularizer: We show that adding a standard weak $L_2$ regularizer on the first decoder layers resolves this ambiguity by penalizing unnecessary shared capacity, guiding the model to the correct $k_s=0$ solution. The regularization helps stabilize the private subspaces and reduces the shared rank, successfully pushing information out of the shared space into the correct modality-specific spaces. We include our results on this edge case in Supplementary Table 8 and Section 4.1.3 lines 376ff.
>
> -------
>
> > **Q1: Purpose of $k_{min}$ and $k_{max}$**
>
> The variables $k_{min}$ and $k_{max}$ (minimum and maximum ranks) define the search bounds for the low-rank decomposition of the weight matrices ($W$). They are crucial for ensuring convergence:They are initialized in Algorithm 1, Line 2, with $k_{min} \leftarrow 1$ and $k_{max} \leftarrow l$ (the initial embedding size).They are dynamically updated (Algorithm lines 12 and 16) to tighten the search space during optimization, preventing the rank from oscillating or moving outside the successfully explored region.
>
> > **Q2: Initial rank values**
>
> The initial values for the ranks are set to the maximum embedding size, $l$, for each respective subspace. This ensures the autoencoder starts with a high capacity during the pre-training phase (Phase 1). Ranks are set to their "previous values" (Line 17) if the fidelity drops below the acceptable threshold, meaning the compression was too aggressive. Since the model begins with the maximum capacity and the highest initial fidelity ($R_{0}^2$), it is highly unlikely for the rank to be increased immediately upon starting Phase 2 (Rank Optimization), but if this would happen, the rank could not be increased beyond the initial size $l$ which is the initial value of $k_{max}$.
>
> > **Q3**
> Thank you for noting the typos. We have proofread the manuscript after adding changes based on the reviews.

---

### Official Review · Reviewer_XWg9 · 2025-10-27

**Soundness:** 2
**Presentation:** 3
**Contribution:** 2
**Rating:** 4
**Confidence:** 3

**Summary:**

This paper focuses on the problem of learning latent representations for multi-modal data. For such a problem, determining the intrinsic dimension is a crucial problem. The authors propose a method to adaptively adjust the intrinsic dimension when learning multi-modal autoencoders. In specific, they multiply the latent representation with a weight matrix $W$, and apply SVD on $W$ every several epochs. The criteria for choosing rank is the reconstruction fidelity. If it is increased, the rank is reduced, otherwise increasing. This rank adaptation process is performed after pre-training of the AE, until convergence of the learned rank. The authors conduct experiments on several simulation datasets and two real datasets to show that the proposed model is able to learn the intrinsic dimensions.

**Strengths:**

The method is simple to use and has potential in many AE architectures, which could be an essential component in different ML models.

**Weaknesses:**

1. Lack of strong baselines. The authors compare with several baselines in simulation, but not for real-world data analysis. For results in Table 2, the authors list the range of estimation varying hyperparameters. The proposed method has the smallest and closest range. But could this be because of inappropriate choices of other methods?

2. In experiments, the authors mainly show the results for estimating the rank. But it is unclear what the practical advantage is, e.g., reconstruction performance or interpretability.

3. Although the authors claim that the proposed method targets multi-modal data, I don’t feel there are specific designs for multi-modal settings. The rank selection procedure is applied on each modality sequentially. Moreover, applying the method on multi-modal settings may require choosing different fidelity measurements for different data domains.

**Questions:**

1. As said in the paper, ARR is one very related paper. ARR applies SVD on latent representations, while this work applies SVD on a projection matrix. What is the motivation of this difference?

2. Did the authors consider adding sparse regularization during either pre-training or rank optimization? Would it help the rank learning?

---

> ### Author Response · Authors · 2025-11-21
> **(1/2)**
>
> We thank the reviewer for their time and for raising several stimulating discussion points, especially regarding the comparison with ARR, which we found to be excellent additions to the paper’s discussion. We also appreciate the reviewer recognizing the simplicity and potential of our approach.
>
> We realize that Algorithm 1 in the initial submission may have been ambiguous regarding the simultaneous nature of the optimization. We are eager to clarify this below to demonstrate the multi-modal design setting and practical advantages of FiGURO.
>
> > **W1: Lack of strong baselines**
>
> We thank the reviewer for the feedback. We are confident that our benchmark on simulated data is both comprehensive and rigorous, reflecting our best efforts to ensure a fair comparison across fundamentally different methodologies.
>
> - **Scope and necessity of simulation:** Simulating data with known ground truth is the only way we can evaluate the accuracy and robustness of our ID estimation method. We have, however, **expanded our benchmarks to multi-modal simulation and real data** based on suggestions of reviewer y7sV. Available multi-modal baselines strongly under-perform in ID estimation and disentanglement on simulated data compared to our method (see Table 2 , Supplementary Figures 4-7, Section 4.1.2 lines 326-328 and Section 4.1.3 lines 369-372). For real data, we also include a comparison to unimodal embeddings and multi-modal baselines on downstream task performance (classification accuracy, see Section 4.2.2 and Table 3).
> - **Fairness:** We tried to give an extensive comparison, including statistical estimators covering all major categories, and neural network approaches. We initially included deterministic approaches only, but have now added a probabilistic approach to the uni-modal benchmark (ARD-VAE) to make it even more robust. We tested baselines across wide hyperparameter ranges where no recommendations were provided, and comparable ranges for similar methods. The smaller ID estimation range reported for FiGURO is not a result of unfair comparison, but a direct demonstration of its superior stability. We have revised former Table 2 (now Supplementary Table 2) to report both the ranges and the best-case estimates. This confirms that while other methods can achieve good results, their extreme sensitivity to hyperparameters is a significant weakness that FiGuRO reliably overcomes.
>
> > **W2: Practical advantages**
>
> While the primary focus of FiGuRO is indeed accurate ID estimation (ranks), the method provides immediate, quantifiable advantages:
> - **Reconstruction:** We added Supplementary Table 12 showing that reconstruction loss barely increases (0.2% on the test set) when reducing dimensions from their initial high values to the final, low estimated ranks. This shows that we can maintain performance while gaining interpretability.
> - **Interpretability:** Our method achieves explicit and stable disentanglement by quantifying the Intrinsic Dimension (ID) for shared and private subspaces. This structural separation of information improves interpretability in the sense that it highlights which information is shared and unique between the modalities (demonstrated in Figures 2B and 3B). This directly translates into practical gains. Our now expanded analysis on real datasets confirms that these interpretable, disentangled subspaces **improve performance in downstream tasks**, such as classification, compared to unimodal embeddings alone (again, see Section 4.2.2 and new Table 3).

---

> ### Author Response · Authors · 2025-11-21
> **(2/2)**
>
> > **W3: Multi-modal design**
>
> We appreciate the reviewer raising this concern, as it allows us to clarify a key aspect of our methodology and improve our communication in the paper. We respectfully confirm that FiGuRO contains crucial, specific designs for multi-modal data, and the rank optimization is explicitly **non-sequential**. The process is a simultaneous, conditional optimization governed by the logic detailed in Algorithm 1. We pre-train the multi-modal autoencoder until early stopping and evaluate the initial fidelity per modality (which allows us to choose **different fidelity metrics per modality**) before starting the rank optimization.
>
> The optimization logic is based on the conditional status of all modalities:
>
> - **Reduction:** All ranks can be reduced only if both modalities' fidelities are within the budget ($\lambda$). If only one modality is satisfied, only that modality's specific rank and the shared rank can be reduced.
> - **Increase:** If all modalities' fidelities have been below the threshold for a defined duration, all subspaces can be increased. If only one modality requires an increase, only its specific rank and the shared rank are increased. If one modality needs an increase and another requires a decrease, the shared space rank is not affected.
> - Because the optimization is simultaneous and conditional, the shared rank $k_s$ only changes if both modalities agree.
>
> We see how this may have been misunderstood due to a lack of detail in **Algorithm 1**, which we have now addressed in the revised manuscript.
>
> --------
>
> > **Q1: ARR vs FiGuRO**
>
> That is a great question. ARR applies SVD on the latent representations and thus can only make **batch estimates** (needing large batch sizes for good approximations), while our approach rather applies SVD to the weights for a global low-rank approximation (similar to LoRA). This allows FiGuRO to operate efficiently without the large batch sizes required by ARR to approximate the covariance matrix. In addition, we modified the SVD to not only give us low-rank approximations, but actual low-dimensional representations by pruning the irrelevant dimensions. We have clarified this distinction now in lines 151-154.
>
> > **Q2: Sparse regularization**
>
> Another good question. We had previously tested the addition of an orthogonality loss (Frobenius norm), which occasionally led to collapsing modalities. We performed **new experiments** adding orthogonality loss and L1 regularization to the latent spaces with varying loss weights. Our results showed that L1 regularization provided no reliable trends in rank estimation, accuracy, or predictability across our multi-modal simulations. Furthermore, the orthogonality loss generally produced no consistent benefit. While it slightly increased the shared rank in one dataset, it led to worse overall disentanglement in $B_s, B_{i1}, B_{i2}$. Please see Section 4.1.3 lines 373-376 in the revised manuscript for details.

---

### Official Review · Reviewer_Gqdy · 2025-10-30

**Soundness:** 2
**Presentation:** 2
**Contribution:** 2
**Rating:** 4
**Confidence:** 3

**Summary:**

Previous dimension estimation methods usually use a unified and static dimension for the representation of the whole dataset, or contrastively estimate for other datasets. This is ill-used for multi-modality data. With this motivation, this paper proposes FiGuRO, which estimates the shared dimension and private dimension for each modality.

**Strengths:**

1. Emphasize the ill-used unified dimension for multi-modality data.
2. Propose FiGuRO, which uses SVD to estimate the dimension and is constrained by the Rate-Distortion Theory.

**Weaknesses:**

1. Some notion is not clear. In equations (3) and (4), SVD is used to estimate the intrinsic dimension. However, this paper did not describe how to estimate the shared dimension $k_s$ and private dimension $k_1$. Instead, the low-rank weight matrices are directly used in equation (4).
2. Experiments are mostly designed in simulation. In real-world data, a good dimension will result in better performance in downstream tasks. More evaluation on downstream tasks will be helpful.
3. The SVD and Rate-Distortion Theory are both commonly used for intrinsic dimension; the method only replicates SVD for multimodality data.

**Questions:**

1. For the loss function, if only R(D) is used?
2. If the hyperparameter distortion threshold $\lambda$ has a special meaning, why only analyze this hyperparameter in Figure 2?
3. In Figure 2, the evaluation metrics accuracy and goodness of R are clear, how to read the rank in Figure 2A? If close to the ground truth (dashed lines) indicates better performance?

---

> ### Author Response · Authors · 2025-11-21
> **(1/2)**
>
> We thank the reviewer for their time and engaging with our work. We appreciate the reviewer’s validation of our core motivation: that a unified, static dimension is often ill-suited for complex multi-modal data. Below, we clarify the specific mechanisms of our shared/private estimation and provide further evidence of performance on downstream tasks to address the concerns regarding real-world utility.
>
> > **W1: Notation**
>
> We see that we may not have been explicit enough in linking the preliminaries to the method introduction. The estimation of the shared ($k_s$) and private ($k_m$) dimensions is not static. It is an iterative process driven by the reconstruction fidelity described in Algorithm 1. To clarify the mechanics:
>
> - **Rank selection:** As introduced in Section 3.1 (last paragraph), the reduced rank $k*$ for any matrix is determined by the number of singular values with cumulative energies above $1 - \gamma$. Equation (3) applies this to our weight matrix $\mathbf{W}$ to obtain a low-rank approximation.
> - **Latent construction and shared/private estimation:** Equation (4) describes the forward pass building on equation (2) from the preliminaries. In the multi-modal setting (for 2 modalities), we have three weight matrices: $\mathbf{W_s}$, $\mathbf{W_1}$, $\mathbf{W_2}$. By multiplying them with concatenated unimodal and individual embeddings ($z_1 \oplus z_2$, $z_1$, $z_2$ respectively), they produce joint and modality-specific embeddings $h$. During training, each of the weight matrices are learned as low-rank approximations independently. This is how shared and private dimensions are obtained.
>
> We **adjusted our methods section** to clarify these core concepts of our approach. Please see the highlighted text (blue) in Section 3.2 for our edits.
>
> > **W2: Real-world data experiments**
>
> We thank the reviewer for the suggestion to include more downstream task evaluation on real data to strengthen our paper. We respectfully emphasize that the extensive use of simulations is necessary because ID estimation requires ground truth IDs to validate the method’s accuracy and stability across varying data characteristics. We agree that downstream task performance can be an important indicator of utility. To address this, we have substantially **broadened our results**. In addition to the Audio MNIST digit prediction on FiGURO’s subspaces alone (accuracies reported in Figure 3B), we now report label classification from each subspace, a **comparison to unimodal embeddings**, and a **comparison against baseline methods** suggested by reviewer y7sV. Our results on Audio MNIST are strong: Comparing concatenated unimodal embeddings to our disentangled embeddings, the **digit classification accuracy increased by 21%** from 0.80 to 0.94. This expanded evidence confirms that the structurally quantified subspaces learned by FiGuRO provide a highly interpretable and effective basis for downstream tasks. Please see Section 4.2.2 for our new results.
>
> > **W3: Methodological contribution**
>
> While SVD and Rate-Distortion Theory (RDT) may be common components of ID estimation, our proposed method is **more than a simple replication of SVD** for multi-modal data and is convincing in its contribution through **practicality and performance**. The core of our work lies in the Fidelity-Guided Rank Optimization (FiGURO) algorithm (**Algorithm 1**), which actively links the two concepts.
>
> - **Dynamic optimization:** Unlike static SVD-based methods, FiGURO uses a Rate-Distortion criterion to dynamically drive the rank estimation. This mechanism allows ranks to both increase and decrease iteratively until convergence, a **crucial difference** from passive rank reduction and even neural network based approaches such as rank reduction autoencoders (RRA).
> - **Empirical evidence for necessity:** To confirm the necessity of this algorithm, we conducted an additional **ablation study** using an SVD-only approach (simple rank reduction as in RRAs but on $\mathbf{W}$) without the RDT-guided decision loop in the multi-modal setting. This naive approach universally collapsed for all multi-modal simulation datasets, underestimating the ranks and not picking up on any of the differences between shared and private information. The new results can be found in Supplementary Table 3 and described in lines 371-373.
>
> Our novelty therefore lies in the fusion of these principles and their efficient implementation into an active, convergent framework for multi-modal ID estimation.

---

> ### Author Response · Authors · 2025-11-21
> **(2/2)**
>
> > **Q1**
>
> The question is formulated ambiguously. If we misinterpreted it, please feel free to clarify the question.
> - Interpretation 1: “Is the training objective equivalent to minimizing the Rate-Distortion function R(D)?” - Training objective and R(D) are not the same. The training objective is often a metric that can take any positive value (e.g. MSE). In order for the distortion budget to be interpretable and comparable among modalities, we prefer metrics with an upper and lower bound, like goodness of fit $R^2$ or explained variance score.
> - Interpretation 2: “Is the reconstruction loss the only loss term?” - Yes, the reconstruction loss is the only loss term. Due to the architectural design and algorithm, reconstruction loss is sufficient (in most cases, see answer to reviewer y7sV for degenerate cases and answer to reviewer XWg9 for additional loss terms) for ID estimation of the different components.
>
> > **Q2: Hyperparameter $\lambda$**
>
> The hyperparameter $\lambda$, which defines the acceptable loss in reconstruction fidelity, was **extensively analyzed** across all simulated experiments in **Section 4.1**, not just Figure 2.
> - We tested its sensitivity on uni-modal data (Section 4.1.1 , Supplementary Table 1 ).
> - We used it as the basis for reporting the range of uni-modal ID estimates in our benchmark (Section 4.1.2, former Table 2 now Supplementary Table 2).
> - Figure 2 presents its effect across four different multi-modal simulation scenarios (Section 4.1.3 ).
>
> Following this comprehensive analysis, we identified a robust default setting ($\lambda=0.05$) for subsequent real-world applications. In our discussion, we note that “For robust estimation, we recommend using FiGuRO to estimate bounds under low and high distortion budgets.” (lines 504f).
>
> > **Q3**
>
> You are correct. In Figure 2A, the estimated rank is considered better the closer it is to the ground truth (GT) dashed lines. Estimates that closely track the GT lines, especially when the lines are far apart (demonstrating successful disentanglement of scale), indicate superior performance. We now explicitly add this clarification to the caption of Figure 2 to ensure it is clear to all readers.

---

> > ### Comment · Reviewer_Gqdy · 2025-11-27
> >
> > Thanks for your response and clarification. For W3, if I understand correctly, the learning of the intrinsic dimension is controlled by two components: (i) the adaptive rank reduction (ARR) via the threshold ($\gamma$), and (ii) the rate–distortion term with distortion budget \($\lambda$\). I am curious: in Algorithm 1, after deciding to increase or decrease the dimensionality for a given modality, by how many dimensions is it adjusted? In other words, how is \($k_{t+1} \leftarrow k_t$\) updated in practice?
> >
> > Moreover, regarding the ablation study in Table 3, does it mean that the rank (or number of singular values) for all modalities is fixed to 1? If so, it seems that with only SVD the estimated dimension is always 1, both for the shared components and for the private components of each modality.

---

> > > ### Author Response · Authors · 2025-11-27
> > >
> > > Thank you for taking the time to read our rebuttal.
> > >
> > > You are absolutely correct, ID learning is controlled by adaptive rank reduction and the rate-distortion metric. In our work, the new rank is chosen as follows in practice:
> > > - Reduction: We compute the cumulative energies of the singular values as in ARR (lines 140-145) and choose the singular values with cumulative energy up to $1 - \gamma$, i.e. $k^* = \max (k \in \{1, \dots, l\} \mid E(k) < 1 - \gamma )$. The unused dimensions are pruned from $h$ (line 153).
> > > - Increase: We keep track of all ranks during our optimization. If the fidelity metric is below $R^2 - \lambda$ and we need to increase the rank, we increase $k_t$ either to the previous rank $k_{t-1}$ or, if that is too large, by 10% (Algorithm 1 line 25 should say “$k_t$ to $\max(k_t + 1, \min(1.1k_t, k_{t-1}))$”, that will be corrected). Increasing the rank reactivates dimensions in $h$.
> > >
> > > In the ablation study, we compared FiGuRO with default hyperparameters (from 4.1.1) with SVD-only optimization. You have correctly observed that excluding the rate-distortion stopping criterion led to an extreme rank reduction, the collapse down to the minimum value of 1. This result highlights the necessity of the rate-distortion optimization loop.
> > >
> > > We hope these explanations were satisfactory and are happy to discuss any additional questions.

---

### Official Review · Reviewer_KrzM · 2025-10-31

**Soundness:** 3
**Presentation:** 3
**Contribution:** 3
**Rating:** 6
**Confidence:** 4

**Summary:**

This paper introduces FiGuRO (Fidelity-Guided Rank Optimization), a framework for intrinsic dimension (ID) estimation in unimodal and multimodal data. FiGuRO combines rate-distortion theory with low-rank projections that adaptively adjust latent dimensionality based on reconstruction fidelity. It separates shared and private subspaces across modalities, providing interpretable and efficient ID estimation. Experiments on synthetic and real data demonstrate accurate and stable results outperforming existing baselines.

**Strengths:**

1.	FiGuRO combines rate-distortion theory with adaptive low-rank learning, providing a principled and theoretically sound approach for intrinsic dimension estimation in multimodal data.

2.	The method is evaluated on various synthetic and real datasets, covering scalar, image, and temporal modalities, and show improvement over baselines in both stability and accuracy.

3.	The idea that the dimensions of different modalities might differ is realistic and novel, providing significance for real-world applications.

4.	The framework can be implemented within standard autoencoders. This simplicity makes FiGuRO highly practical and broadly applicable across different data types and model settings, enhancing its potential for adoption in real-world multimodal learning pipelines.

**Weaknesses:**

1.	Some implementation details (e.g., how the distortion budget $\gamma$ or threshold $\lambda$ is set) are not fully discussed, which may affect reproducibility. Also, the sensitivity regarding these hyperparameters is not evaluated.

**Questions:**

Is the method sensitive to the hyperparameters? And is there any experimental validation in this aspect?

---

> ### Author Response · Authors · 2025-11-21
>
> We thank the reviewer for their encouraging feedback and for recognizing FiGURO as a “principled and theoretically sound approach”. We particularly appreciate the reviewer highlighting our thorough evaluation and practical significance. Below we address the concerns and questions regarding implementation details and hyperparameter sensitivity analysis.
>
> > **W1 + Q1**
>
> **Implementation details**
>
> The distortion budget $\lambda$ is a hyperparameter of the method. For distortion metrics with fixed ranges ($R^2$, Explained Variance) which we mainly focus on, this defines the acceptable decrease in that metric and thus the threshold. The distortion threshold is $R^2_{0} - \lambda$. We introduce $\lambda$ in Section 3.2 lines 168f and subsequently explain the fidelity metric $R^2$ and how $R^2_{0} - \lambda$ is used to determine when to stop reduction and potentially increase dimensions as a fidelity threshold (lines 170-174). Algorithm 1 further shows how $\lambda$ and $R^2_{0} - \lambda$ are used (algorithm lines 12,13).
>
> The other threshold used in this paper is the rank reduction threshold $\gamma$. As briefly mentioned in the preliminaries in Section 3.1 (line 145), $\gamma$ is the threshold for the cumulative energy of the singular values in SVD used by RRAs as the primary hyperparameter for reduction. In our hyperparameter sensitivity analysis, we observed that $\gamma$ had a negligible effect on our predicted ranks, as the main driver of when to stop in our method is $\lambda$. Thus, as mentioned in Section 4.1.1, we use $\gamma = 0.01$ in all our experiments. We realize that this insight was not properly communicated in the initial draft of the paper and **have clarified this choice now** in lines 288-290 and Algorithm 1 line 1, and renamed $\gamma$ the “energy threshold” to be more precise and avoid confusion. We have also **added details to Algorithm 1** based on feedback from multiple reviewers. We would also like to note that with publication, we will make both source code and all experiment scripts publicly available for reproducibility.
>
> **Hyperparameter sensitivity**
>
> Section 4.1.1 contains a **sweep over all method-specific hyperparameters** ($\lambda, \tau, \gamma, \pi$). As mentioned in the section, the overall ID prediction of 4.89 with a ground truth of 5 and a standard error of the mean of 0.01 (N=1080) showed that the method was very robust to hyperparameters, with all detailed predictions given in Supplementary Table 1. The distortion budget $\lambda$ was the only hyperparameter showing a small but clear dependency with the predicted rank. Given that this trend turned out to be more pronounced in the multimodal setting (Figure 2A), we see now that our discussion of the hyperparameter sensitivity was lacking this observation and was thus not adequate. We have now elaborated on this in lines 288-290.
>
> We hope these clarifications strengthen your assessment and are happy to answer any further questions.

---

### Author Response · Authors · 2025-11-26
**General Response**

We sincerely thank all reviewers for the time, effort, and high-quality feedback provided during this discussion period. Your constructive questions and insights have led to a substantially strengthened manuscript and enhanced clarity regarding the methodology.

### **Core contributions recognized by reviewers**

We are highly encouraged by the positive feedback provided by several reviewers, which consistently validated the motivation, novelty, and practical potential of FiGuRO.
To summarize, the reviewers found our work to be:
- **Principled and theoretically sound**: Reviewer KrzM recognized FiGuRO as a "principled and theoretically sound approach for intrinsic dimension estimation in multimodal data" and noted its thorough evaluation.
- **Novel methodology**: Reviewer y7sV highlighted the "novel methodology" and found the proposed method of greedily adapting the latent dimension rank to achieve the desired error "justified".
- **Highly practical and applicable**: Reviewer KrzM praised the framework's simplicity, noting its potential for "broad adoption in real-world multimodal learning pipelines". Reviewer XWg9 similarly noted the method is "simple to use and has potential in many AE architectures". Reviewer y7sV further noted the paper's "Clear motivation and writing" strong.

### **Summary of key changes and clarifications**

In response to the reviewers’ comments, we have clarified methodological ambiguities and performed significant updates and new experimental analyses, resulting in a substantially strengthened manuscript. The related changes have been highlighted in blue in the revised manuscript.
1. Clarifying **methodological novelty** (To reviewers Gqdy, XWg9, y7sV):
    - We corrected the misunderstanding that our method is a simple sequential application of SVD. FiGuRO employs a **simultaneous, conditional rank optimization logic designed specifically for multi-modal data** (Algorithm 1).
    - We performed an **ablation study** showing that a naive "SVD-only" approach universally collapses in multi-modal settings, proving that our Fidelity-Guided Rank Optimization algorithm is the **critical, novel component** (new Supplementary Table 3).
    - We clarified that FiGuRO **not only estimates ranks but also learns the interpretable subspaces themselves**.
2. **Expanded baselines and real-world utility** (To reviewers Gqdy and y7sV):
    - We performed **new comparisons against multi-view decomposition methods**. Our results show that while linear baselines fail by swapping shared/private information under certain signal conditions, FiGuRO maintains robust disentanglement (Section 4.1.3, Supplementary Figures 4-7) and achieves the best ID estimates (new Table 2).
    - We addressed the insufficient empirical illustration of utility by expanding our **real-world downstream task analysis**. Our results demonstrate that the **disentangled subspaces improve classification performance** compared to unimodal embeddings and baselines, empirically validating FiGuRO's practical advantage (Section 4.2.2).

### **Conclusion**

We once again thank all the reviewers for their valuable time and insightful feedback. We believe that the extensive experimental additions and detailed clarifications address all major concerns. We are confident that FiGuRO is a **robust, novel, and practical contribution** that **fills a critical gap in multi-modal representation learning**. We encourage all reviewers to re-evaluate their scores based on the strength and depth of our revised manuscript and discussion. We are happy to answer any further questions.

---

### Meta-Review · Area_Chair_mteM · 2026-01-03

**Summary:**

Based on the reviewer’s comments, the authors’ responses, and my own assessment, I believe the paper would benefit from an additional round of peer review rather than being accepted in its current form. My reasons are as follows:

1. In their rebuttal, the authors have acknowledged the validity of the reviewers’ concerns and have made substantial technical revisions to the manuscript—for instance, Algorithm 1 has been significantly modified. These extensive changes indicate that the paper is not yet ready for publication in its present state.

2. In my view, the problem space of the paper remains inadequately defined. The notion of intrinsic data dimensionality is inherently contingent on the type of model used to fit the data. For example, the estimated intrinsic dimension may increase when employing a linear subspace model compared to a nonlinear manifold. Without clarifying the underlying modeling assumptions, the investigation of intrinsic dimension estimation appears insufficiently grounded.

**Reviewer Scores:**

NA

---

### Decision · Program_Chairs · 2026-01-26

Reject